# Estimating Dimensionality of Neural Representations from Finite Samples

**Chanwoo Chun**[1,2][*][†]    **Abdulkadir Canatar**[3][*]    **SueYeon Chung**[1,3]    **Daniel Lee**[3,4]

[1] Harvard University    [2] New York University    [3] Flatiron Institute    [4] Cornell Tech

## Abstract

The global dimensionality of a neural representation manifold provides rich insight into the computational process underlying both artificial and biological neural networks. However, all existing measures of global dimensionality are sensitive to the number of samples, i.e., the number of rows and columns of the sample matrix. We show that, in particular, the participation ratio of eigenvalues, a popular measure of global dimensionality, is highly biased with small sample sizes, and propose a bias-corrected estimator that is more accurate with finite samples and with noise. On synthetic data examples, we demonstrate that our estimator can recover the true known dimensionality. We apply our estimator to neural brain recordings, including calcium imaging, electrophysiological recordings, and fMRI data, and to the neural activations in a large language model, and show that our estimator is invariant to the sample size. Finally, our estimators can additionally be used to measure the local dimensionalities of curved neural manifolds by weighting the finite samples appropriately.

## 1 Introduction

How does a population of a million neurons encode an input or stimulus? This question is central in neuroscience and machine learning (ML). In a standard geometric view, the population response to a stimulus is a vector in a high-dimensional space whose axes are individual neurons' activation levels. Varying the stimulus forms a set of representations, a neural manifold, in this space. A basic question in this framework is: what is the dimensionality of that manifold? Although fundamental, many dimensionality estimators are sensitive to sample size (the number of stimuli and recorded neurons) and to measurement noise. Local (intrinsic) dimensionality estimators exist that are invariant to sample size, such as the TwoNN method, but they cannot measure global dimensionality, and they are often highly sensitive to noise (Facco et al., 2017; Denti et al., 2022). Despite the long history and proven utility of global dimensionality in neuroscience and ML, there is no estimator that is resistant to both finite sample size and noise (Mineault et al., 2024). We aim to close this gap.

Global dimensionality, understood as an effective rank given by the number of nonzero or effectively nonzero singular values of a data matrix, provides rich insights. It has been used to understand computation in brains and deep networks, to quantify classification (Chung et al., 2018; Cohen et al., 2020; Sorscher et al., 2022) and regression performance (Zhang, 2002; Caponnetto and De Vito, 2007; Bach, 2013), to train linear probes in artificial neural networks (Shah et al., 2025), and to design brain-computer interface (BCI) decoders (Willett et al., 2021; Menendez et al., 2025; Willsey et al., 2025). Cohen et al. (2020), Chou et al. (2025), and Sorscher et al. (2022) relate global dimensionality to the linear separability of manifolds and observe that dimensionality tends to decrease across successive processing stages in both the convolutional neural network and visual cortex, which improves linear separability. Estimating global dimensionality is also useful for interpretability research in large language models (LLMs). For example, linear probes applied to intermediate LLM layers can reliably classify harmful versus non-harmful content (Shah et al., 2025; Kantamneni et al., 2025; Smith et al., 2025), so tracking global dimensionality across layers yields important insights for AI safety and interpretability (Shah et al., 2025). In BCI, it is a standard practice to estimate

---

[*]Equal contribution.
[†]Work done while at Weill Cornell Medical College.

the global dimensionality of neural activations to first understand the encoding scheme of the motor cortex and utilize this insight to guide the design of BCI decoders (Willett et al., 2021; Menendez et al., 2025; Willsey et al., 2025). Beyond these applications, analyzing global dimensionality has long been a staple in neuroscience, ML, genomics (Pocrnic et al., 2016), and behavior science (Woo et al., 2023) for characterizing collective behavior in large systems.

Global dimensionality estimated from finite data is systematically biased. In neuroscience, the observed activation matrix has shape $P \times Q$ (stimuli × neurons) and is effectively a random submatrix of a much larger, unobserved matrix. One can present only a small subset of stimuli and record only a subset of neurons. Counting large singular values of the observed matrix, or of its covariance, is highly sensitive to the available numbers of rows and columns, a widely noted challenge across many fields (Woo et al., 2023; Pocrnic et al., 2016; Mazzucato et al., 2016; Mineault et al., 2024). Common workarounds include subsampling to produce saturation curves that are checked visually as the matrix approaches the full dataset size (Woo et al., 2023), and ad-hoc extrapolation when saturation is not observed (Lehky et al., 2014).

We address this problem with a principled estimation-theoretic approach. We correct the finite-sample bias of a widely used global dimensionality metric, the participation ratio (PR) of covariance eigenvalues. The PR is a soft count of nonzero singular values and is widely used in neuroscience and machine learning (Meissner-Bernard et al., 2024; Fortunato et al., 2024; Harvey et al., 2024; Beiran et al., 2023; Niemeyer et al., 2022; Susman et al., 2021; Gao et al., 2017; Rajan et al., 2010; Kirsanov et al., 2025; Schrage et al., 2024; Kuoch et al., 2024; Yerxa et al., 2023; Sorscher et al., 2022; Fort et al., 2022; Mel and Ganguli, 2021; Cohen et al., 2020; Harvey et al., 2024). However, just like other global dimensionality estimators, the existing PR estimators exhibit substantial finite-sample bias (Mazzucato et al., 2016; Menendez et al., 2025). Previously, there have been non-rigorous attempts at correcting the bias (Dahmen et al., 2020; Pospisil and Pillow, 2024), but they require strong assumptions on distributions, only account for row sampling, and provide biased estimators of the numerator and denominator of PR. Here, we present a rigorous bias-corrected estimator of PR by deriving the unbiased estimators of the numerator and denominator that require only very weak assumptions and account for both row and column sampling. We also provide an extension that removes the bias contributed by noise, performs sampling correction via importance sampling, is applicable to sparse matrices, and a variation that measures local dimensionality, which is resistant to noise, unlike the existing popular local dimensionality estimator, TwoNN (Denti et al., 2022).

## 2 Definitions and Problem setup

Here, we provide an informal formulation of the problem setup. A more rigorous and thorough problem formulation is presented in the Supplementary section Sec. A.

### 2.1 Representation matrix

In neuroscientific and ML experiments, the neural representation data takes the form of a matrix $\Phi \in \mathbb{R}^{P \times Q}$, where each row represents an input (stimulus), and each column represents a single feature (neuron). We assume that the neural activation $\Phi_{i\alpha}$ is given by a hypothetical map $\phi$ that maps an input $x_i$ and a parameter $w_\alpha$ that parameterizes the neuron:

$$\Phi_{i\alpha} = \phi(x_i, w_\alpha) \tag{1}$$

As it will be evident later, our dimensionality estimator is agnostic of $\phi$ and the distributions of $x_i$ and $w_\alpha$. Nonetheless, we define this generative process to establish a clear mental framework.

### 2.2 Participation ratio

We are interested in measuring the dimensionality of the activation matrix in the limit where the number of neurons and inputs approach infinity. Let us denote the matrix in this limit as $\Phi^{(\infty)} \in \mathbb{R}^{P^{(\infty)} \times Q^{(\infty)}}$ where $P^{(\infty)}, Q^{(\infty)} \to \infty$. A dimensionality measure should quantify how many non-zero eigenvalues the covariance matrix $\mathbf{K}^{(\infty)} := \frac{1}{Q^{(\infty)}} \Phi^{(\infty)} \Phi^{(\infty)\top}$ has. In the most strict sense, this is given by the rank of $\mathbf{K}^{(\infty)}$. However, the rank is sensitive to small eigenvalues that one might want

to ignore. Therefore, a softer count of eigenvalues, participation ratio (PR) $\gamma$, has become a popular measure of dimensionality, which is defined as the following:

$$\gamma_0 := \frac{\left(\sum_i \lambda_i\right)^2}{\sum_i \lambda_i^2}$$

where $\{\lambda_i\}$ is the set of all eigenvalues of $\mathbf{K}^{(\infty)}$. Suppose $D$ number of the eigenvalues all take a single value $c$, and the rest are all zero. In this case, $\gamma_0$ is $D$, which equals the matrix rank. However, if there is an additional small eigenvalue $\epsilon$, then the PR changes only by a small amount $\gamma_0 \approx D + \mathcal{O}(\epsilon)$, whereas the matrix rank abruptly becomes $D + 1$. Therefore $\gamma_0$ forms a lower bound on the matrix rank. It has been shown that $\gamma_0$ number of the largest eigenvalues typically explain 70–80% of the variance in the data (Gao et al., 2017).

We introduce equivalent ways of expressing the PR, which will be useful in the later sections. We use the fact that the sum of the eigenvalues of a positive semi-definite matrix is given by its trace, and the sum of the squares of the eigenvalues is given by the trace of its square:

$$\gamma_0 \equiv \frac{\frac{1}{P^{(\infty)2}}\mathrm{tr}\left(\mathbf{K}^{(\infty)}\right)^2}{\frac{1}{P^{(\infty)2}}\mathrm{tr}\left(\mathbf{K}^{(\infty)2}\right)} \equiv \frac{\left\langle \bar{v}_{iijj}^{\alpha\beta}\right\rangle}{\left\langle \bar{v}_{ijij}^{\alpha\beta}\right\rangle}, \quad \text{where} \quad \bar{v}_{ijkl}^{\alpha\beta} := \Phi_{i\alpha}^{(\infty)}\Phi_{j\alpha}^{(\infty)}\Phi_{k\beta}^{(\infty)}\Phi_{l\beta}^{(\infty)} \tag{2}$$

and $\langle\cdot\rangle$ is a notation for averaging over all free indices, e.g.

$$\left\langle \bar{v}_{ijlr}^{\alpha\beta}\right\rangle = \frac{1}{\text{\# of summands}} \sum_{ijlr}\sum_{\alpha\beta} \bar{v}_{ijlr}^{\alpha\beta}.$$

The second equality in Eqn. 2 is obtained by simply expanding the matrix notation in terms of the activations, e.g. $\Phi_{i\alpha}^{(\infty)} \equiv \phi(x_i, w_\alpha)$.

We consider the case where each column (neuron) is centered before computing the dimensionality, which is a common practice in neuroscience and ML. Consider a neural manifold in $\mathbb{R}^{Q^{(\infty)}}$ where each point in the manifold is a row vector of $\Phi^{(\infty)}$. This centering operation simply shifts the manifold such that its center of mass is at the origin. The dimensionality $\gamma$ of the centered manifold is given by

$$\gamma := \frac{A}{B} \tag{3}$$

$$\text{where} \quad A := \left\langle \bar{v}_{iijj}^{\alpha\beta}\right\rangle - 2\left\langle \bar{v}_{iijl}^{\alpha\beta}\right\rangle + \left\langle \bar{v}_{ijlr}^{\alpha\beta}\right\rangle, \quad \text{and} \quad B := \left\langle \bar{v}_{ijij}^{\alpha\beta}\right\rangle - 2\left\langle \bar{v}_{ijjl}^{\alpha\beta}\right\rangle + \left\langle \bar{v}_{ijlr}^{\alpha\beta}\right\rangle.$$

Note that we use the notation $\gamma_0$ to refer to the uncentered dimensionality, and $\gamma$ to refer to the centered dimensionality. We study $\gamma$ for the rest of the paper. The centering above (subtracting each column's mean, i.e. centering across stimuli) gives rise to the *task dimensionality*, which measures the dimensionality of the stimulus-driven neural variance. An alternative centering of each row (subtracting the mean across neurons for each stimulus) gives the *neuron dimensionality*. The two are generally not equal; we use task centering throughout the main paper. See Sec. A for precise definitions and guidance on which to use.

## 2.3 SAMPLE MATRIX AND NAIVE ESTIMATOR

The sample activation matrix $\Phi \in \mathbb{R}^{P \times Q}$ is a random submatrix of $\Phi^{(\infty)}$, obtained by selecting the $P$ rows and $Q$ columns of $\Phi^{(\infty)}$ independently and uniformly at random, and then collecting the entries at the intersections of the selected rows and columns. In most neuroscience experiments, one can only observe a subset of neurons in a given brain region, which corresponds to the column sampling. In both neuroscience and ML experiments, one can only present a subset of stimuli (e.g., it is impossible to present all possible natural images), which corresponds to the row sampling. In terms of the generative process defined in Eqn. 1, the sampling of the submatrix is equivalent to independently drawing $P$ stimuli $\{x_i\}_{i=1}^P$ and $Q$ neuronal parameters $\{w_\alpha\}_{\alpha=1}^Q$ from some distributions $\rho_{\mathcal{X}}$ and $\rho_{\mathcal{W}}$, respectively, i.e. $x_i \sim \rho_{\mathcal{X}}$, and $w_\alpha \sim \rho_{\mathcal{W}}$.

In the literature, the PR is commonly estimated by simply substituting $\Phi^{(\infty)}$ with $\Phi$ in Eqn. 3:

$$\gamma_{\text{naive}} := \frac{A_{\text{naive}}}{B_{\text{naive}}}, \tag{4}$$

where $\quad A_{\text{naive}} := \left\langle v_{iijj}^{\alpha\beta} \right\rangle - 2 \left\langle v_{iijl}^{\alpha\beta} \right\rangle + \left\langle v_{ijlr}^{\alpha\beta} \right\rangle, \quad B_{\text{naive}} := \left\langle v_{ijij}^{\alpha\beta} \right\rangle - 2 \left\langle v_{ijjl}^{\alpha\beta} \right\rangle + \left\langle v_{ijlr}^{\alpha\beta} \right\rangle,$

and $\quad v_{ijkl}^{\alpha\beta} := \Phi_{i\alpha}\Phi_{j\alpha}\Phi_{k\beta}\Phi_{l\beta}.$

Note that the only difference between Eqn. 3 and Eqn. 4 is that the Eqn. 4 is computed on the submatrix $\Phi$, whereas Eqn. 3 is computed on the true matrix $\Phi^{(\infty)}$. However, this naive estimator is very sensitive to the number of observed neurons $Q$ and stimuli presented $P$. In the next sections, we identify the source of the sample-size-sensitive bias in this estimator, and then propose our estimator that corrects the bias.

## 3 BEHAVIOR OF NAIVE ESTIMATOR

### 3.1 SCALING LAW OF DIMENSIONALITY OF SAMPLE MATRIX

We find that the naive PR estimate, i.e., the dimensionality of the sample matrix, scales as

$$\mathbb{E}_\Phi \left[ \frac{1}{\gamma_{\text{naive}}} \right] \approx \frac{1}{P} + \frac{1}{Q} + \frac{1}{\gamma},$$

under the assumptions that the norms of the rows (and also columns) of $\Phi^{(\infty)}$ are uniform and centered, and $P$, $Q$, and $\gamma$ are large (see Sec. B for details). This provides a remarkably intuitive insight: $\gamma_{\text{naive}}$ is approximately a harmonic mean of $P$, $Q$, and $\gamma$, up to a constant factor, much like the law of parallel resistance. An example generative process that can satisfy these assumptions is $\phi(x, w) = x^\top G w$ where $G$ is some constant matrix with large dimensionality, $x$ and $w$ are isotropic random vectors with finite variance.

### 3.2 BIAS IN NAIVE ESTIMATOR

Although the simple substitution in $\gamma_{\text{naive}}$ is intuitive, it leads to a heavily biased estimation of $\gamma$. This is because both the numerator and denominator of $\gamma_{\text{naive}}$ are biased estimates of the numerator and denominator of $\gamma$: $\mathbb{E}_\Phi [A_{\text{naive}}] \neq A$ and $\mathbb{E}_\Phi [B_{\text{naive}}] \neq B$ where $\mathbb{E}_\Phi [\cdot]$ denotes the average over the independent uniform sampling of the submatrix. In fact, each term in $A_{\text{naive}}$ (or $B_{\text{naive}}$) is a biased estimate of a corresponding term in $A$ (or $B$). For example, $\mathbb{E}_\Phi \left[ \left\langle v_{iijj}^{\alpha\beta} \right\rangle \right] \neq \left\langle \bar{v}_{iijj}^{\alpha\beta} \right\rangle$, which are the first terms of $A_{\text{naive}}$ and $A$. If we write this inequality explicitly in terms of the matrix entries $\Phi_{i\alpha} \equiv \phi(x_i, w_\alpha)$, we have

$$\mathbb{E}_\Phi \left[ \frac{1}{P^2 Q^2} \sum_{ij} \sum_{\alpha\beta} \phi(x_i, w_\alpha)^2 \phi(x_j, w_\beta)^2 \right] \neq \mathbb{E}_{x,w} \left[ \phi(x, w)^2 \right]^2,$$

where $\mathbb{E}_{x,w} [\cdot]$ denotes the average over the distributions of $x$ and $w$. Consider decomposing the sum in the LHS into the terms where none of the indices coincide, and the rest:

$$\frac{1}{P^2 Q^2} \mathbb{E}_\Phi \left[ \sum_{i \neq j} \sum_{\alpha \neq \beta} \phi(x_i, w_\alpha)^2 \phi(x_j, w_\beta)^2 + \text{rest} \right]$$

Moving the expectation inside the sum over unequal indices, for the first term, we have $\mathbb{E}_\Phi \left[ \phi(x_i, w_\alpha)^2 \phi(x_j, w_\beta)^2 \right]$ for $i \neq j$ and $\alpha \neq \beta$. However, since the row sampling and column sampling are both independent, $\phi(x_i, \cdot)$ and $\phi(x_j, \cdot)$ are independent for $i \neq j$, and $\phi(\cdot, w_\alpha)$ and $\phi(\cdot, w_\beta)$ are also independent for $\alpha \neq \beta$. Therefore, the first term factorizes to $\mathbb{E}_\Phi \left[ \phi(x_i, w_\alpha)^2 \right] \mathbb{E}_\Phi \left[ \phi(x_j, w_\beta)^2 \right]$, which is simply $\mathbb{E}_{x,w} \left[ \phi(x, w)^2 \right]^2$, the quantity we want to estimate. However, the "rest" term is non-zero, i.e. $\mathbb{E}_\Phi [\text{rest}] \neq 0$, contributing to the bias. Note that in the "rest" term, the indices in $\phi(x_i, w_\alpha)^2 \phi(x_j, w_\beta)^2$ are not all unequal, making $\phi(x_i, w_\alpha)^2$ and $\phi(x_j, w_\beta)^2$ correlated, so the expectation cannot be factorized, contributing as bias.

## 4 BIAS-CORRECTED GLOBAL DIMENSIONALITY ESTIMATOR

Having identified in the previous section that overlapping indices in a sum contribute to the bias in each term of $A_{\text{naive}}$ (or $B_{\text{naive}}$), we now know that unbiased estimators of $A$ and $B$ can be found by simply averaging over unequal indices. Let us first define a notation for averaging over unequal indices for both rows and columns:

$$\left\langle v_{ijlr}^{\alpha\beta} \right\rangle_{\text{both}} = \frac{1}{\text{\# of summands}} \sum_{i,j,l,r}^{\#} \sum_{\alpha,\beta}^{\#} v_{ijlr}^{\alpha\beta}. \tag{5}$$

$\sum^{\#}$ denotes a summation restricted to mutually distinct values of the free indices. For example, $\sum_{i,j,l}^{\#}$ excludes all summands where any pair among $i$, $j$, and $l$ coincides. When the subscript pattern forces certain indices to be equal, as in $\left\langle v_{ijij}^{\alpha\beta} \right\rangle_{\text{both}}$, the summation runs over the free indices only, here $\sum_{i,j}^{\#}$. Then, our unbiased estimators of $A$ and $B$ are

$$A_{\text{both}} := \left\langle v_{iijj}^{\alpha\beta} \right\rangle_{\text{both}} - 2 \left\langle v_{iijl}^{\alpha\beta} \right\rangle_{\text{both}} + \left\langle v_{ijlr}^{\alpha\beta} \right\rangle_{\text{both}}, \quad \text{and}$$

$$B_{\text{both}} := \left\langle v_{ijij}^{\alpha\beta} \right\rangle_{\text{both}} - 2 \left\langle v_{ijjl}^{\alpha\beta} \right\rangle_{\text{both}} + \left\langle v_{ijlr}^{\alpha\beta} \right\rangle_{\text{both}}.$$

Finally, we define our estimator for the true dimensionality as simply the ratio of $A_{\text{both}}$ and $B_{\text{both}}$:

$$\gamma_{\text{both}} := \frac{A_{\text{both}}}{B_{\text{both}}}.$$

Note that even if $\hat{X}$ and $\hat{Y}$ are unbiased estimators of $X$ and $Y$, $\hat{X}/\hat{Y}$ is a biased estimate of $X/Y$. The ratio operation introduces a small but inevitable bias, which cannot be reduced further in a straightforward manner (see Sec. D). In general, however, the bias contributed by the ratio operation is negligible compared to that contributed by biases in the numerator and denominator. We provide a detailed theoretical analysis of the bias and variance of these estimators in Sec. D.

If one desires, one can correct only the bias contributed from row sampling by summing over unequal row indices, but still summing over all column indices. This could be useful if one has full observation of neurons, but the inputs are sampled. In a similar manner, one can only correct the bias contributed by column sampling. We refer to these estimators as $\gamma_{\text{row}}$ and $\gamma_{\text{col}}$, respectively.

### 4.1 IMPLEMENTATION

The main challenge in implementation is in performing the sum over unequal indices. Vectorizing or parallelizing the sum is highly non-trivial. Consider, for example, a sum of the following form, $\sum_{i,j,k}^{\#} u_{ijk}$ for some tensor $u$. To vectorize this sum, one needs to express it in terms of a regular sum that sums over all indices. After some algebra, the unequal sum can be expanded as:

$$\sum_{i,j,k}^{\#} u_{ijk} \equiv \sum_{ijk} u_{ijk} - \sum_{ij} u_{iij} - \sum_{ij} u_{ijj} - \sum_{ij} u_{iji} + 2 \sum_{i} u_{iii}$$

where now the vectorized calculation is possible for each sum on a computer. One can simply use the `einsum` subroutine for each sum. In our case, we have four row indices $\{i, j, l, r\}$ and two column indices $\{\alpha, \beta\}$, each set required to be mutually distinct, yielding six sets of constraints. This makes the expanded forms considerably longer than the 3-index example above; the complete, implementable expressions for all estimators ($\gamma_{\text{both}}$, $\gamma_{\text{row}}$, and $\gamma_{\text{col}}$) are provided in Sec. A.

### 4.2 EXTENSIONS

#### 4.2.1 NOISE CORRECTION

In many scenarios, the sample matrix is corrupted by an additive or multiplicative noise. This is inevitable in neural recordings. We show that we can correct the bias from the additive or multiplicative noise (or both simultaneously), as long as two sample matrices are obtained over two trials for fixed sets of stimuli and neurons. An alternative naive approach would be to perform $N$ trials

and take an element-wise average of the $N$ sample matrices, before passing it to the dimensionality estimator. However, performing multiple trials is typically expensive, and the bias contributed by the noise would be $\mathcal{O}\left(1/\sqrt{N}\right)$ with this naive method. In contrast, our method, inspired by Stringer et al. (2019), only requires $N = 2$ trials, and the bias contributed by the noise is $\mathcal{O}\left(1/P + 1/Q\right)$, much more efficient than the alternative method. There, we assume the sample matrix in $t$-th trial is generated by

$$\Phi_{i\alpha}^{(t)} = \phi(x_i, w_\alpha) + \eta(x_i, w_\alpha, \epsilon_t)$$

where $\epsilon_t$ is sampled independently across trial, and $\mathbb{E}_\epsilon[\eta(x, w, \epsilon)] = 0$. If $\eta(x_i, w_\alpha, \epsilon_t) = \epsilon_t \phi(x_i, w_\alpha)$ and $\epsilon_t$ is zero-mean, then this models multiplicative noise: $\Phi_{i\alpha}^{(t)} = (1 + \epsilon_t)\, \phi(x_i, w_\alpha)$.

To correct for the bias due to noise, we can simply redefine $v_{ijkl}^{\alpha\beta}$ as $v_{ijkl}^{\alpha\beta} \leftarrow \Phi_{i\alpha}^{(1)}\Phi_{j\alpha}^{(2)}\Phi_{k\beta}^{(1)}\Phi_{l\beta}^{(2)}$ where $\Phi^{(1)}$ and $\Phi^{(2)}$ are the sample matrices from two trials. The rest of the calculation can be performed as explained in the previous section to obtain the dimensionality estimates. See Sec. A for the formal derivation, Sec. D for its bias-variance analysis, and Sec. A.4 for a discussion of which noise structures are neutralized by this construction, and the $N$-trial extension for correlated noise.

### 4.2.2 Importance sampling and local dimensionality

In real life, the distribution of neurons $\rho_\mathcal{W}$ in a given brain region could be different from the distribution of the observed neurons $\rho_\mathcal{W}^{\mathrm{obs}}$. Similarly, the distribution of the intended set of stimuli $\rho_\mathcal{X}$ may be different from that of the available set of stimuli $\rho_\mathcal{X}^{\mathrm{obs}}$. In this case, importance-sampling (IS) weights $r(x) = \rho_\mathcal{X}(x)/\rho_\mathcal{X}^{\mathrm{obs}}(x)$ and $c(w) = \rho_\mathcal{W}(w)/\rho_\mathcal{W}^{\mathrm{obs}}(w)$ can be incorporated into the unequal-index sums to obtain the bias-corrected estimator $\gamma_{\mathrm{both}}^{\mathrm{IS}}$; the full derivation and a concrete V1 example are given in Sec. A.5.

There exists a simpler, yet less rigorous, implementation. Suppose $\mathcal{S} = \{s_i\}_{i=1}^P$ is a set of scaling factors, each of which determines how much we want to weight a given neural response. We now define a weighted sum notation:

$$\left\langle v_{ijlr}^{\alpha\beta} \right\rangle_{\mathrm{both}}^{\mathcal{S}} := \frac{\sum_{i,j,l,r}^{\#}\sum_{\alpha,\beta}^{\#} s_i s_j s_l s_r v_{ijlr}^{\alpha\beta}}{Q\left(Q-1\right)\sum_{i,j,l,r}^{\#} s_i s_j s_l s_r}.$$

Each free row index contributes exactly one scaling factor. For instance, $\left\langle v_{ijij}^{\alpha\beta} \right\rangle_{\mathrm{both}}^{\mathcal{S}}$ has two free row indices $i$ and $j$, so the numerator contains $s_i s_j v_{ijij}^{\alpha\beta}$ and the denominator normalizes by $Q(Q-1)\sum_{i,j}^{\#} s_i s_j$. Then the weighted dimensionality is given by

$$\gamma_{\mathrm{both}}^{\mathcal{S}} := \frac{\left\langle v_{iijj}^{\alpha\beta} \right\rangle_{\mathrm{both}}^{\mathcal{S}} - 2\left\langle v_{iijl}^{\alpha\beta} \right\rangle_{\mathrm{both}}^{\mathcal{S}} + \left\langle v_{ijlr}^{\alpha\beta} \right\rangle_{\mathrm{both}}^{\mathcal{S}}}{\left\langle v_{ijij}^{\alpha\beta} \right\rangle_{\mathrm{both}}^{\mathcal{S}} - 2\left\langle v_{ijjl}^{\alpha\beta} \right\rangle_{\mathrm{both}}^{\mathcal{S}} + \left\langle v_{ijlr}^{\alpha\beta} \right\rangle_{\mathrm{both}}^{\mathcal{S}}}.$$

One can also estimate the local dimensionality, i.e., intrinsic dimensionality, around a given point in the representation space by assigning large weights to the neighboring points and small weights to distant points. In conjunction with the noise correction method explained above, our local dimensionality estimator is resistant to the additive/multiplicative noise, unlike popular local dimensionality estimators (see Sec. 5).

### 4.2.3 Estimating from sparse sample matrix

Sometimes, the observed entries of an underlying matrix do not form a complete matrix due to an unstructured sparsity, e.g., missing neurons for certain trials in neural recording, and a real-life user-item matrix in a recommender system. If we precisely define the "# of summands" in Equation (5) as the number of summands that do not include any missing entry, we can ensure that the $A_{\mathrm{both}}$ and $B_{\mathrm{both}}$ are unbiased estimators of $A$ and $B$, even when the sample matrix is sparse. However, one should assume that the occurrence of a missing entry and the sampling of $x$ or $w$ are independent.

#### 4.2.4 ESTIMATING FINITE-SIZE UNDERLYING MATRIX

In some cases, the underlying matrix from which a submatrix is sampled may not be infinitely large, but of finite size, $R \times C$, where $R \geq P$ and $C \geq Q$. In this case, the rows and columns of $\Phi$ are sampled without replacement from the finite underlying matrix $G$. This makes the row (and also column) sampling process no longer independent, requiring an alternative formulation of the estimators. We present the unbiased estimators of the numerator and denominator of the PR of $G$ in the Sec. A.6. These estimators require the knowledge of $R$ and $C$.

### 4.3 SYNTHETIC DATA

We first verify our estimator on synthetic data by testing how quickly different estimators converge to their true dimensionality. Here, we considered the following noisy linear generative process:

$$\Phi_{i\alpha} = x_i \cdot w_\alpha + \epsilon_{i\alpha} \tag{6}$$

We sample $Q$ feature variables $\{w_\alpha\}_{\alpha=1}^Q$ and $P$ inputs $\{x_i\}_{i=1}^P$ independently from $\mathcal{N}(0, \mathbf{I}_d)$, and the noise term $\{\epsilon_{i\alpha}\}$ from $\mathcal{N}(0, \sigma_\epsilon^2)$ to form a $P \times Q$ sample matrix $\Phi$. In the limit when $P, Q \to \infty$, the true, noise-free, PR should approach $d$, which we refer to as the true dimensionality $\gamma$ for this setup. Note that a finite size $\Phi$ obtained by the above process can be seen as a random, noisy submatrix of an underlying infinite matrix.

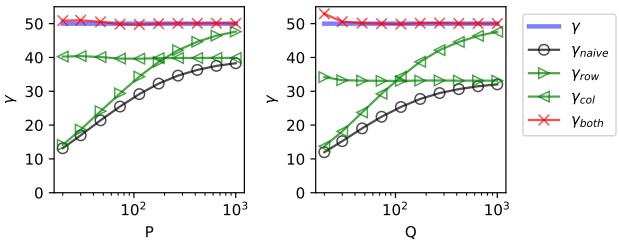

Figure 1: Different dimensionality estimates of the linear model with $d = 50$ and noise variance $\sigma_\epsilon^2 = 0.2$. **Left:** varying number of stimuli $P$ with fixed $Q = 100$ neurons. **Right:** varying number of neurons $Q$ with fixed $P = 200$ stimuli. The horizontal line marks the true dimensionality $\gamma = d = 50$.

We find that our estimator $\gamma_{\text{both}}$ is able to recover the true dimensionality across wide ranges of finite $P$ and $Q$ (Figure 1). Note that our estimator did not require any information about the distributions of $x$ and $w$, or the map $\phi$. We do observe a very small bias in our estimator when $P$ or $Q$ is very small, due to the nonlinear effect of taking the ratio of unbiased estimators, as described earlier in Section 4. However, this bias is negligible compared to the bias in the other estimators.

### 4.4 BRAIN DATA

In real data scenarios, the true dimensionality $\gamma$ is unknown, and we can only assess the estimators' performance based on how quickly they converge to the dimensionality calculated using the entire dataset. Here, we test our estimator on multiple neural datasets, all using natural image stimuli:

1. Mouse V1 recorded with calcium imaging Stringer et al. (2019)
2. Macaque V4 recorded with microelectrode arrays (local field potential) Papale et al. (2025)
3. Macaque IT recorded with microelectrode arrays (spike-sorted) Majaj et al. (2015)
4. Human IT fMRI data Hebart et al. (2023)

By subsampling from each dataset, we vary the number of neuronal units[1] $Q$ or the number of natural images $P$, and apply the dimensionality estimators. In Figure 2, we report our results as a function of the number of subsamples sampled from the full dataset.

---

[1]They are either ROIs (calcium imaging), spiking units (Ephys), electrodes (Ephys), or voxels (fMRI).

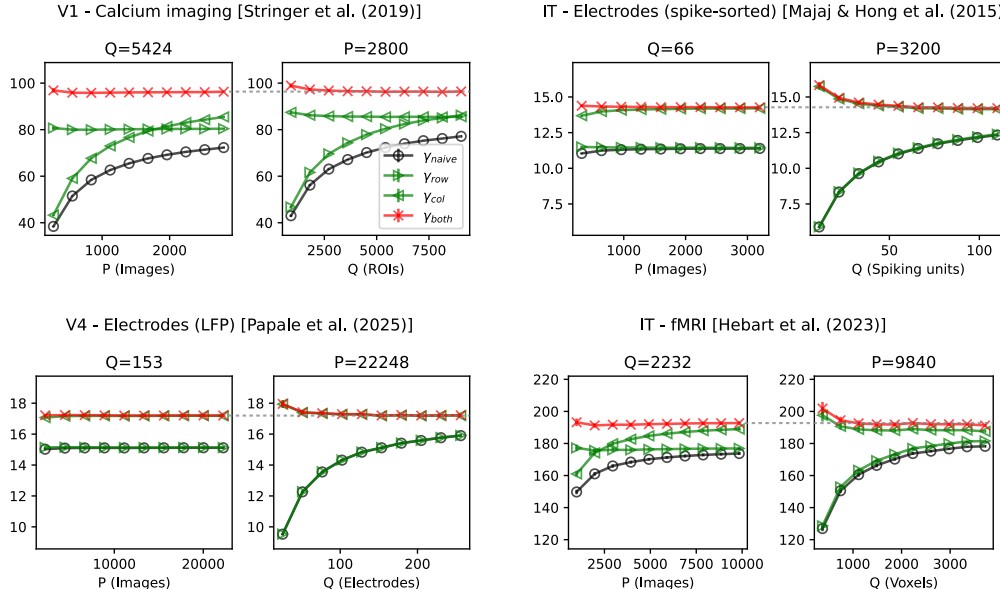

Figure 2: Dimensionality estimates on four different neural recording datasets for varying number of stimuli $P$, and neural activation units $Q$, by subsampling from the full dataset. Top left: Mouse V1 (Stringer et al., 2019); Top right: Macaque IT (Majaj et al., 2015); Bottom left: Macaque V4 (Papale et al., 2025); Bottom right: Human IT (Hebart et al., 2023). Note that the estimators converge to *different* plateau values: $\gamma_{\text{naive}}$ retains residual bias from both $P$ and $Q$ sampling; $\gamma_{\text{row}}$ (or $\gamma_{\text{col}}$) corrects only row (or column) sampling bias and so plateaus at an intermediate value; $\gamma_{\text{both}}$ remains remarkably constant across $P$ and $Q$ variations (grey dotted line extends $\gamma_{\text{both}}$ at the max $P$).

Most notably, the empirical mean of $\gamma_{\text{both}}$ is consistently least sensitive to the number of samples, compared to that of the other estimators (Figure 2). Note that when varying the number of rows $P$, the $\gamma_{\text{row}}$ that corrects the bias due to row sampling is practically invariant, but it is sensitive to $Q$ (Figure 2). The opposite is true for $\gamma_{\text{col}}$. Finally, $\gamma_{\text{naive}}$ is sensitive to both $P$ and $Q$ and is the most biased. These results indicate that our estimator is useful regardless of neural recording modality and can capture the underlying dimensionality with a relatively much smaller number of samples.

## 4.5 ARTIFICIAL NEURAL NETWORKS

We also apply our estimator to artificial neural networks. In this case, one has access to the entire population of neurons; hence, correcting for overlapping column indices should not make any difference. However, there may be input-limited cases where only a few exemplars of a particular class can be accessed.

Here, we consider this case for evaluating the dimensionality of hidden layers of large language models (LLM). We use the FLORES+ dataset NLLB Team et al. (2024) for multilingual machine translation and extract representations from the hidden layers of a pretrained Llama3 base model Grattafiori et al. (2024). The FLORES+ dataset contains 483 sentences translated to over 200 languages. Here, we picked 9 languages (see Sec. C for details) and extracted their hidden layer representations from the LLM. Since sentences have different lengths, we use the representation of the last token of each sentence. In Fig. 3, we report the average dimensionality across all languages against the input sampling ratio.

Since we fix the number of neurons and subsample the inputs only, we expect column correction to have a tiny effect compared to row correction. In Fig. 3a, we indeed observe that $\gamma_{\text{col}}$ performs as poorly as the naive estimator and that $\gamma_{\text{row}}$ performs as well as our estimator.

Next, we show how the average dimensionality changes from layer to layer in Fig. 3b when the sample size is small. While the naive estimator significantly underestimates the dimensionality, it

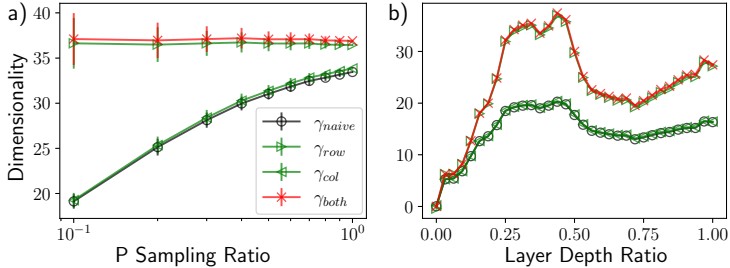

Figure 3: Estimating the task dimensionality of LLM features for different languages. **a)** We calculate the dimensionality of the last layer for each language separately and report its average as a function of the input sampling ratio. In this example, all layers have $Q = 4096$ dimensional representations, and each language has a total of $P = 483$ sentences. The error bars represent the standard deviation for 50 random draws. **b)** The dimensionality profile across layers when the sampling ratio is 0.1.

preserves the overall profile of dimensionality across layers. However, our estimator reveals more fine-grained features of the layerwise dimensionality that are hidden otherwise. We provide additional details about our experiments in Sec. C.

Finally, we would like to comment on the interesting behavior in Fig. 3b, where the dimensionality increases towards the mid-layers and decreases again. This behavior was observed previously Valeriani et al. (2023) and was recently reported in Skean et al. (2025) by using matrix-based entropy measures Giraldo et al. (2014), which includes the logarithm of the dimensionality as a special case.

## 5 EXTENSION: LOCAL DIMENSIONALITY ESTIMATION

In the earlier section, we described the procedure of weighing samples based on importance. Here we demonstrate that one can extend this framework to measure local dimensionality of a manifold, adopting a method inspired by Recanatesi et al. (2022). From a given data point $\overrightarrow{\Phi}_0$ (an arbitrary row vector of $\Phi$), one can measure the distance to the rest of the data points in a manifold dataset. One can then discard the data points that are further than some predefined distance $r$, by giving them zero weights and giving the points inside the local ball radius of $r$ uniform weights. The resulting weighted dimensionality estimate is denoted $\gamma_{\text{both}}^{\text{local}}\left(\overrightarrow{\Phi}_0, r\right)$. One can obtain the average local dimensionality of a given manifold by taking the average of $\gamma_{\text{both}}^{\text{local}}\left(\overrightarrow{\Phi}_0, r\right)$ over all available $\overrightarrow{\Phi}_0$'s:

$$\gamma_{\text{both}}^{\text{local}}(r) := \frac{1}{P} \sum_{i=1}^{P} \gamma_{\text{both}}^{\text{local}}\left(\overrightarrow{\Phi_{i,:}}, r\right)$$

where $\overrightarrow{\Phi_{i,:}}$ denotes $i$th row vector of $\Phi$. In our experiments, we use the Mahalanobis distance with a local metric. See Sec. E for a formal definition of the local dimensionality estimator and distance metric. One can similarly define $\gamma_{\text{naive}}^{\text{local}}(r)$, $\gamma_{\text{row}}^{\text{local}}(r)$, and $\gamma_{\text{col}}^{\text{local}}(r)$.

To confirm that our estimator can indeed capture the true local dimensionality, we test it on a synthetic dataset. The synthetic dataset is created using a random Fourier feature (RFF) model whose covariance kernel converges to the radial basis function (RBF) kernel in the limit where the number of features (neurons) approaches infinity. A noisy RFF model is defined as

$$\phi(x, \{w, b\}) = \sin(x \cdot w + b)$$

where $w \in \mathbb{R}^d$ is sampled independently from a normal distribution $\mathcal{N}(0, \mathbf{I}_d)$, $b$ is sampled independently from a uniform distribution $\mathcal{U}\left(-\pi/2, \pi/2\right)$. The data matrix $\Phi \in \mathbb{R}^{P \times Q}$ is created by presenting $P$ number of inputs ($x$'s) sampled from $\mathcal{N}\left(0, \sigma_x^2 \mathbf{I}_d\right)$ to $Q$ number of random features, such that for a trial $t$, $\Phi_{i\alpha}^{(t)} = \phi(x_i, \{w_\alpha, b_\alpha\}) + \epsilon_{i,\alpha,t}$ where $\epsilon_{i,\alpha,t}$ is the noise term. We emphasize that the noise term $\epsilon_{i,\alpha,t}$ is sampled independently across all $i$, $\alpha$, and $t$ from $\mathcal{N}\left(0, \sigma_\epsilon^2\right)$. In the limit of infinite $P$ and $Q$, the true noise-free local dimensionality of the $Q$-dimensional representation manifold should approach $d$.

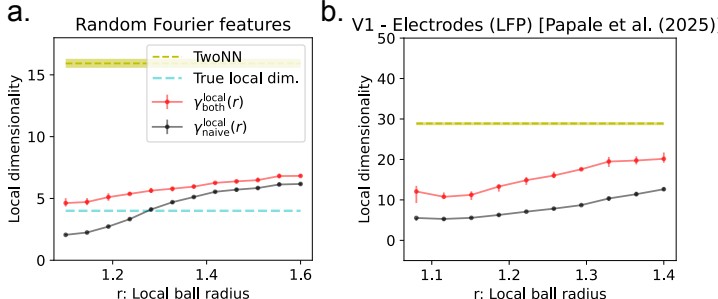

Figure 4: **a**. Estimating the local dimensionality of the random Fourier feature model using TwoNN, $\gamma_{\text{naive}}^{\text{local}}(r)$, and $\gamma_{\text{both}}^{\text{local}}(r)$, while varying the radius of the local ball for the latter two estimators. Signal-to-noise ratio is approximately $3.33$ ($\sigma_\epsilon = 0.3$). **b.** Estimating the local dimensionality of the macaque V1 LFP measured with electrode arrays (Papale et al., 2025).

We find that our local dimensionality measure $\gamma_{\text{both}}^{\text{local}}(r)$ recovers the true local dimensionality $d$ despite the presence of noise as we decrease the local ball radius to the smallest allowable length (Figure 4). The smallest radius is determined by requiring that at least four data points lie inside a local ball. Recall that our estimator requires at least four distinct indices, so four is the minimum allowed dataset size. In contrast, the widely adopted local dimensionality estimator TwoNN significantly overestimates the local dimensionality due to its susceptibility to noise. The naive local estimator $\gamma_{\text{naive}}^{\text{local}}(r)$ also fails to recover the true dimensionality, since it underestimates when there are only a small number of samples in a local ball. This highlights the need to correct for bias due to the small sample size, especially when measuring local dimensionality.

We also apply the local dimensionality estimators to the real brain dataset, V1 LFP measurements from macaque (Papale et al., 2025). We find that the estimate from our estimator $\gamma_{\text{both}}^{\text{local}}(r)$ in the small radius limit is much smaller than the TwoNN estimate (Figure 4).

## 6 Discussions

In this paper, we resolve the timely issue of the sensitivity of the global dimensionality estimator to the sample size and measurement noise. To this end, we correct the bias in the PR, a popular measure of global dimensionality. We then apply our estimators to a synthetic dataset, real neuronal datasets of various recording modalities, and a large language model representation, to show that our estimator is indeed resistant to the sample size. As extensions, we also show how we can measure local dimensionality using our dimensionality estimators.

The PR characterizes the eigenvalue spectrum based on the first and second order statistics of the spectrum. To recover more information about the spectrum, one can estimate the higher spectral moments using the same basic principles employed here (Chun et al., 2025).

## 7 Code Availability

We provide our implementation in https://github.com/badooki/dimensionality.

## 8 Limitations

For measuring local dimensionality, our estimator $\gamma_{\text{both}}^{\text{local}}(r)$ requires computing pairwise distances, which dominates at $\mathcal{O}(P^2 Q)$ time and $\mathcal{O}(P^2 + PQ)$ memory; sweeping over $r$ radii brings the total to $\mathcal{O}(rP^2 Q)$ time and $\mathcal{O}(rP(P+Q))$ memory. This is more expensive than TwoNN, though the computation can be easily parallelized over multiple threads. The global estimator $\gamma_{\text{both}}$ shares the same asymptotic time complexity as the naive estimator: $\mathcal{O}(\min(P, Q)^2 \cdot \max(P, Q))$.

ACKNOWLEDGEMENTS

We thank Jonathan D. Victor for the discussion. Partial support for this work was provided by the Simons Foundation. DDL was also supported by the Institute of Information & Communications Technology Planning & Evaluation (IITP) grant funded by the Korean Government (MSIT) (No. RS-2024-00457882, National AI Research Lab Project). S.C. was partially supported by a Sloan Research Fellowship, a Klingenstein-Simons Award, and the Samsung Advanced Institute of Technology project, "Next Generation Deep Learning: From Pattern Recognition to AI." All experiments were performed using the Flatiron Institute's high-performance computing cluster.

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

# A    DERIVATION OF BIAS CORRECTED ESTIMATORS OF DIMENSIONALITY

## A.1    DEFINITIONS

### KERNEL INTEGRAL OPERATOR

We assume there are latent variables, $x$ and $w$, associated with stimulus and neuron, respectively. To model the sampling of stimulus, we assume there is a distribution $\rho_\mathcal{X}$ over the set $\mathcal{X}$ of all stimulus latent variables. Similarly, to model the sampling of neurons, we assume there is a distribution $\rho_\mathcal{W}$ over the set $\mathcal{W}$ of all neuron latent variables. Then, we assume the entry of the sample matrix is given by a map $\phi : \mathcal{X} \times \mathcal{W} \to \mathbb{R}$

$$\Phi_{i\alpha} = \phi(x_i, w_\alpha). \tag{S1}$$

We let $P$ and $Q$ denote the number of sampled stimuli and neurons in this paper, i.e. $\Phi \in \mathbb{R}^{P \times Q}$. Assume $\phi$ is square-integrable with respect to $\rho_\mathcal{X}$ and $\rho_\mathcal{W}$. Now, define the associated kernel functions

$$k(x, x') := \int d\rho_\mathcal{W}(w)\, \phi(x, w)\phi(x', w), \quad \text{and} \tag{S2}$$

$$\tilde{k}(w, w') := \int d\rho_\mathcal{X}(x)\, \phi(x, w)\phi(x, w'). \tag{S3}$$

There is a kernel integral operator that is associated to $k$, $T_k : \mathcal{L}^2(\rho_\mathcal{X}, \mathcal{X}) \to \mathcal{L}^2(\rho_\mathcal{X}, \mathcal{X})$, defined as

$$T_k f = \int d\rho_\mathcal{X}(x)\, k(\cdot, x)f(x). \tag{S4}$$

$T_k$ is a trace operator and is analogous to the population covariance matrix $K_{\text{all}}$ introduced earlier. The tuple $(\phi, \rho_\mathcal{X}, \rho_\mathcal{W})$ uniquely defines a generative process of data.

### PARTICIPATION RATIO

The eigenvalues $\{\lambda_i\}_{i=1}^\infty$ of $T_k$ are implicitly defined by

$$T_k e_i = \lambda_i e_i \quad \forall i \tag{S5}$$

where $e_i \in \mathcal{L}^2(\rho_\mathcal{X}, \mathcal{X})$, and $\{e_i\}_{i=1}^\infty$ forms an orthonormal set. There are a countably infinite number of eigenvalues. Then, the participation ratio of these eigenvalues is defined as

$$\gamma_0 := \frac{\left(\sum_{i=1}^\infty \lambda_i\right)^2}{\sum_{i=1}^\infty \lambda_i^2} \equiv \frac{\left(\int d\rho_\mathcal{X}(x)\, k(x, x)\right)^2}{\int d\rho_\mathcal{X}(x)d\rho_\mathcal{X}(y)\, k(x, y)^2}. \tag{S6}$$

We refer to $\gamma$ as the true effective dimensionality of $T_k$. If $n$ number of $\lambda_i$'s take a constant value $c$ and the rest are $0$, then the participation ratio is $n$, matching the definition of rank. However, if we decrease one of the positive eigenvalues to be smaller than $c$ (but still positive), then the participation ratio is between $n-1$ and $n$, reflecting the fact that the dimensionality is effectively slightly less than $n$. Therefore, the participation ratio forms a lower bound on the rank, and is "softer" than the rank.

Typically, the participation ratio is computed on the eigenvalues $\{\eta_i\}$ of a sample covariance matrix $K := \frac{1}{Q}\Phi\Phi^\top$:

$$\frac{\text{tr}(K)^2}{\text{tr}(K^2)} \equiv \frac{\left(\sum_i \eta_i\right)^2}{\sum_i \eta_i^2}. \tag{S7}$$

## A.2    CENTERED KERNEL

Suppose $k_c(x, y)$ is a centered kernel, where activations for each neuron is centered:

$$k_c(x_i, x_j) = \int d\rho_\mathcal{W}(w)\, \phi_s(x, w)\phi_s(y, w)$$

where

$$\phi_s(x, w) = \phi(x, w) - \int d\rho_\mathcal{X}(z)\, \phi(z, w).$$

Similarly, $\tilde{k}_c(x, y)$ is another centered kernel, where activations for each stimulus is centered:

$$\tilde{k}_c(x_i, x_j) = \int d\rho_{\mathcal{W}}(w)\, \phi_f(x, w)\phi_f(y, w)$$

where

$$\phi_f(x, w) = \phi(x, w) - \int d\rho_{\mathcal{W}}(z)\, \phi(x, z).$$

Before centering, it did not matter whether we computed the dimensionality with $k$ or $\tilde{k}$:

$$\gamma := \frac{\left(\int d\rho_{\mathcal{X}}(x)\, k(x, x)\right)^2}{\int d\rho_{\mathcal{X}}(x)d\rho_{\mathcal{X}}(y)\, k(x, y)^2} \equiv \frac{\left(\int d\rho_{\mathcal{W}}(w)\, \tilde{k}(w, w)\right)^2}{\int d\rho_{\mathcal{W}}(w)d\rho_{\mathcal{W}}(u)\, \tilde{k}(w, u)^2}.$$

Not only the ratio, but also the numerators and denominators of the two expressions match respectively. However, the centered dimensionalities are different

$$\gamma^{\text{task}} \neq \gamma^{\text{neuron}}, \quad \text{where}$$

$$\gamma^{\text{task}} = \frac{\left(\int d\rho_{\mathcal{X}}(x)\, k_c(x, x)\right)^2}{\int d\rho_{\mathcal{X}}(x)d\rho_{\mathcal{X}}(y)\, k_c(x, y)^2}, \quad \text{and} \quad \gamma^{\text{neuron}} = \frac{\left(\int d\rho_{\mathcal{W}}(w)\, \tilde{k}_c(w, w)\right)^2}{\int d\rho_{\mathcal{W}}(w)d\rho_{\mathcal{W}}(u)\, \tilde{k}_c(w, u)^2}.$$

**Guidance on centering.** The two centered dimensionalities measure conceptually different quantities. *Task dimensionality* $\gamma^{\text{task}}$ (row centering via $k_c$) subtracts from each neuron's response its mean across stimuli, thereby measuring the dimensionality of the *stimulus-driven* variance in neural space. This is the default throughout the main paper and is the natural choice when the scientific question concerns how the stimulus set is coded. *Neuron dimensionality* $\gamma^{\text{neuron}}$ (column centering via $\tilde{k}_c$) subtracts from each stimulus representation its mean across neurons, measuring the dimensionality of the neuron-response patterns. This is more appropriate when the question concerns the structure of neural covariation rather than stimulus coding. In general $\gamma^{\text{task}} \neq \gamma^{\text{neuron}}$; the two coincide only when the data matrix has a special symmetric structure (e.g., when $k = \tilde{k}$ and the row and column distributions are identical). The bias-corrected estimators $\gamma^{\text{task}}_{\text{both}}$ and $\gamma^{\text{neuron}}_{\text{both}}$ are derived identically to $\gamma_{\text{both}}$, with the respective centering applied before computing $v^{\alpha\beta}_{ijlr}$.

For now, let us focus on the numerator and the denominator of $\gamma^{\text{task}}$. Note that the numerator can be expressed in terms of the original kernel $k$ as:

$$A := \left(\int d\rho_{\mathcal{X}}(x)\, k_c(x, x)\right)^2$$

$$= \left(\int d\rho_{\mathcal{X}}(x)\, k(x, x)\right)^2 - 2\int d\rho_{\mathcal{X}}(x)\, k(x, x)\int d\rho_{\mathcal{X}}(x)d\rho_{\mathcal{X}}(y)\, k(x, y) + \left(\int d\rho_{\mathcal{X}}(x)d\rho_{\mathcal{X}}(y)\, k(x, y)\right)^2$$

The denominator can be expanded as:

$$B := \int d\rho_{\mathcal{X}}(x)d\rho_{\mathcal{X}}(y)\, k_c(x, y)^2$$

$$= \int d\rho_{\mathcal{X}}(x)d\rho_{\mathcal{X}}(y)\, k(x, y)^2 - 2\int d\rho_{\mathcal{X}}(x)d\rho_{\mathcal{X}}(y)d\rho_{\mathcal{X}}(z)\, k(x, z)k(z, y) + \left(\int d\rho_{\mathcal{X}}(x)d\rho_{\mathcal{X}}(y)\, k(x, y)\right)^2$$

### A.3 DERIVATION OF THE ESTIMATORS

The goal is to derive an unbiased estimator of each one of the six terms (five unique terms) above. The five unique terms are

$$\bar{t}^1 := \left( \int d\rho_{\mathcal{X}}(x) \, k(x,x) \right)^2, \quad \bar{t}^2 := \int d\rho_{\mathcal{X}}(x) \, k(x,x) \int d\rho_{\mathcal{X}}(x) d\rho_{\mathcal{X}}(y) \, k(x,y),$$

$$\bar{t}^3 := \int d\rho_{\mathcal{X}}(x) d\rho_{\mathcal{X}}(y) \, k(x,y)^2, \quad \bar{t}^4 := \int d\rho_{\mathcal{X}}(x) d\rho_{\mathcal{X}}(y) d\rho_{\mathcal{X}}(z) \, k(x,z) k(z,y),$$

$$\text{and} \quad \bar{t}^5 := \left( \int d\rho_{\mathcal{X}}(x) d\rho_{\mathcal{X}}(y) \, k(x,y) \right)^2.$$

With these terms, the numerator of $\gamma^{\text{task}}$ is

$$A = \bar{t}^1 - 2\bar{t}^2 + \bar{t}^5,$$

and the denominator is

$$B = \bar{t}^3 - 2\bar{t}^4 + \bar{t}^5.$$

Consider $\bar{t}^1$ as an example. We start with deriving a naive biased estimator, since it is easiest.

$$t^1_{\text{naive}} := \frac{1}{P^2 Q^2} \sum_{ij} \sum_{\alpha\beta} \phi(x_i, w_\alpha)^2 \phi(x_j, w_\beta)^2$$

We arrived at the above by simple derivation:

$$P^2 Q^2 t^1_{\text{naive}} := \left( \text{Tr}(K) \right)^2 = \left( \sum_i \sum_\alpha \Phi_{i\alpha}^2 \right)^2 \tag{S8}$$

$$= \sum_{ij} \sum_{\alpha\beta} \Phi_{i\alpha}^2 \Phi_{j\beta}^2 = \sum_{ij} \sum_{\alpha\beta} \Phi_{i\alpha}^2 \Phi_{j\beta}^2 = \sum_{ij} \sum_{\alpha\beta} \phi(x_i, w_\alpha)^2 \phi(x_j, w_\beta)^2. \tag{S9}$$

The reason $t^1_{\text{naive}}$ is biased, i.e. $\bar{t}^1 \neq \left\langle t^1_{\text{naive}} \right\rangle_\Phi$ where $\langle \cdot \rangle_\Phi$ is the average over all submatrices, is the following:

$$\left\langle t^1_{\text{naive}} \right\rangle_\Phi = \left\langle \frac{1}{P^2 Q^2} \sum_{ij} \sum_{\alpha\beta} \phi(x_i, w_\alpha)^2 \phi(x_j, w_\beta)^2 \right\rangle_{\{x_i\},\{w_\alpha\}}$$

$$= \frac{1}{P^2 Q^2} \left( \sum_{i \neq j} \sum_{\alpha \neq \beta} \left\langle \phi(x_i, w_\alpha)^2 \phi(x_j, w_\beta)^2 \right\rangle + \sum_i \sum_{\alpha \neq \beta} \left\langle \phi(x_i, w_\alpha)^2 \phi(x_i, w_\beta)^2 \right\rangle \right.$$

$$\left. + \sum_{i \neq j} \sum_\alpha \left\langle \phi(x_i, w_\alpha)^2 \phi(x_j, w_\alpha)^2 \right\rangle + \sum_i \sum_\alpha \left\langle \phi(x_i, w_\alpha)^4 \right\rangle \right)$$

$$= \frac{(P-1)(Q-1)}{PQ} \left\langle k(x,x) \right\rangle_{x \sim \rho_{\mathcal{X}}}^2 + \frac{Q-1}{PQ} \left\langle k(x,x)^2 \right\rangle_{x \sim \rho_{\mathcal{X}}}$$

$$+ \frac{P-1}{PQ} \left\langle \tilde{k}(w,w)^2 \right\rangle_{w \sim \rho_{\mathcal{W}}} + \frac{1}{PQ} \left\langle \phi(x,w)^4 \right\rangle_{x \sim \rho_{\mathcal{X}}, w \sim \rho_{\mathcal{W}}} \tag{S10}$$

Note that in the first term, $\left\langle k(x,x) \right\rangle_{x \sim \rho_{\mathcal{X}}}^2 = \bar{t}^1$, so the leading order term in $\left\langle t^1_{\text{naive}} \right\rangle_\Phi$ is unbiased. However, the rest of the terms contribute to $\mathcal{O}\left( \frac{1}{P} + \frac{1}{Q} \right)$ bias. This shows that we can derive the

unbiased estimator by simply summing over disjoint indices, and normalizing by the number of summands:

$$t_{\text{both}}^1 = \frac{1}{P(P-1)Q(Q-1)} \sum_{i \neq j} \sum_{\alpha \neq \beta} \phi(x_i, w_\alpha)^2 \phi(x_j, w_\beta)^2.$$

Then we have $\langle t_{\text{both}}^1 \rangle_\Phi = \bar{t}^1$. Applying the same logic to all five terms $\bar{t}^1$, $\bar{t}^2$, $\bar{t}^3$, $\bar{t}^4$, and $\bar{t}^5$, we arrive at the following unbiased estimators:

$$t_{\text{both}}^1 = \frac{1}{P(P-1)Q(Q-1)} \sum_{i \neq j} \sum_{\alpha \neq \beta} v_{iijj}^{\alpha\beta},$$

$$t_{\text{both}}^2 = \frac{1}{P(P-1)(P-2)Q(Q-1)} \sum_{i \neq j \neq l} \sum_{\alpha \neq \beta} v_{iijl}^{\alpha\beta},$$

$$t_{\text{both}}^3 = \frac{1}{P(P-1)Q(Q-1)} \sum_{i \neq j} \sum_{\alpha \neq \beta} v_{ijij}^{\alpha\beta},$$

$$t_{\text{both}}^4 = \frac{1}{P(P-1)(P-2)Q(Q-1)} \sum_{i \neq j \neq l} \sum_{\alpha \neq \beta} v_{ijjl}^{\alpha\beta}, \quad \text{and}$$

$$t_{\text{both}}^5 = \frac{1}{P(P-1)(P-2)(P-3)Q(Q-1)} \sum_{i \neq j \neq l \neq r} \sum_{\alpha \neq \beta} v_{ijlr}^{\alpha\beta},$$

where

$$v_{ijlr}^{\alpha\beta} := \Phi_{i\alpha} \Phi_{j\alpha} \Phi_{l\beta} \Phi_{r\beta}.$$

The remaining challenge is computing the sums over disjoint indices. The implementation is challenging, since computing this sum over a loop can be slow on a computer. Therefore, one needs to re-express a disjoint sum into a linear combination of regular sums. For example:

$$\sum_{i \neq j} r_{ij} = \sum_{i,j} r_{ij} - \sum_i r_i.$$

This becomes non-trivial as the number of indices increases. Consider an example with three indices:

$$\sum_{i \neq j \neq k} r_{ijk} = \sum_{i,j,k} r_{ijk} \left(1 - \delta_{ij}\right) \left(1 - \delta_{ik}\right) \left(1 - \delta_{jk}\right)$$

$$= \sum_{i,j,k} r_{ijk} \left(1 - \delta_{jk} - \delta_{ik} - \delta_{ij} + 2\delta_{ijk}\right)$$

$$= \sum_{i,j,k} r_{ijk} - \sum_{i,j} r_{ijj} - \sum_{i,j} r_{iji} - \sum_{i,j} r_{iij} + 2 \sum_i r_{iii}.$$

Using this technique, we find that the terms can be expanded into the following. Let

$$r_{ijlr} := \frac{1}{Q(Q-1)} \sum_{\alpha \neq \beta} v_{ijlr}^{\alpha\beta} = \frac{1}{Q(Q-1)} \left( \sum_{\alpha,\beta} v_{ijlr}^{\alpha\beta} - \sum_\alpha v_{ijlr}^{\alpha\alpha} \right)$$

then

$$t_{\text{both}}^1 = \frac{\sum_{ij} r_{iijj} - \sum_i r_{iiii}}{P(P-1)},$$

$$t_{\text{both}}^2 = \frac{\sum_{ijl} r_{iijl} - 2\sum_{ij} r_{ijjj} - \sum_{ij} r_{iijj} + 2\sum_i r_{iiii}}{P(P-1)(P-2)},$$

$$t_{\text{both}}^3 = \frac{\sum_{ij} r_{ijij} - \sum_i r_{iiii}}{P(P-1)},$$

$$t_{\text{both}}^4 = \frac{\sum_{ijl} r_{ijjl} - 2\sum_{ij} r_{iiij} - \sum_{ij} r_{ijij} + \sum_i 2r_{iiii}}{P(P-1)(P-2)}, \quad \text{and}$$

$$t_{\text{both}}^5 = \frac{\sum_{ijlm} r_{ijlm} - 2\sum_{ijl}\left(r_{iijl} + 2r_{ijjl}\right) + \sum_{ij}\left(r_{iijj} + 8r_{ijjj} + 2r_{ijij}\right) - 6\sum_i r_{iiii}}{P(P-1)(P-2)(P-3)}.$$

Finally, the unbiased estimators of the numerator and the denominator of $\gamma_{\text{task}}$ are

$$A_{\text{both}} = t_{\text{both}}^1 - 2t_{\text{both}}^2 + t_{\text{both}}^5, \quad \text{and}$$

$$B_{\text{both}} = t_{\text{both}}^3 - 2t_{\text{both}}^4 + t_{\text{both}}^5.$$

By simply taking the ratio of these unbiased estimators, we obtain our dimensionality estimator

$$\gamma_{\text{both}} = \frac{A_{\text{both}}}{B_{\text{both}}}.$$

Deriving $A_{\text{row}}$ and $B_{\text{row}}$ can be achieved by simply summing over the columns with the regular sum, but the rows with disjoint indices: Let

$$r'_{ijlr} := \frac{1}{Q^2} \sum_{\alpha\beta} v_{ijlr}^{\alpha\beta}$$

Then,

$$t_{\text{row}}^1 = \frac{\sum_{ij} r'_{iijj} - \sum_i r'_{iiii}}{P(P-1)},$$

$$t_{\text{row}}^2 = \frac{\sum_{ijl} r'_{iijl} - 2\sum_{ij} r'_{ijjj} - \sum_{ij} r'_{iijj} + 2\sum_i r'_{iiii}}{P(P-1)(P-2)},$$

$$t_{\text{row}}^3 = \frac{\sum_{ij} r'_{ijij} - \sum_i r'_{iiii}}{P(P-1)},$$

$$t_{\text{row}}^4 = \frac{\sum_{ijl} r'_{ijjl} - 2\sum_{ij} r'_{iiij} - \sum_{ij} r'_{ijij} + \sum_i 2r'_{iiii}}{P(P-1)(P-2)}, \quad \text{and}$$

$$t_{\text{row}}^5 = \frac{\sum_{ijlm} r'_{ijlm} - 2\sum_{ijl}\left(r'_{iijl} + 2r'_{ijjl}\right) + \sum_{ij}\left(r'_{iijj} + 8r'_{ijjj} + 2r'_{ijij}\right) - 6\sum_i r'_{iiii}}{P(P-1)(P-2)(P-3)}.$$

Then the row-corrected estimators are given by

$$A_{\text{row}} = t_{\text{row}}^1 - 2t_{\text{row}}^2 + t_{\text{row}}^5,$$

$$B_{\text{row}} = t_{\text{row}}^3 - 2t_{\text{row}}^4 + t_{\text{row}}^5, \quad \text{and}$$

$$\gamma_{\text{row}} = \frac{A_{\text{row}}}{B_{\text{row}}}.$$

Similarly, $A_{\text{col}}$ and $B_{\text{col}}$ can be achieved by simply summing over the rows with the regular sum, but the columns with disjoint indices:

$$t_{\text{col}}^1 = \frac{1}{P^2} \sum_{i,j} r_{iijj},$$

$$t_{\text{col}}^2 = \frac{1}{P^3} \sum_{i,j,l} r_{iijl},$$

$$t_{\text{col}}^3 = \frac{1}{P^2} \sum_{i,j} r_{ijij},$$

$$t_{\text{col}}^4 = \frac{1}{P^3} \sum_{i,j,l} r_{ijjl}, \quad \text{and}$$

$$t_{\text{col}}^5 = \frac{1}{P^4} \sum_{i,j,l,r} r_{ijlr}.$$

Note that the unbiased estimators for $\gamma_{\text{neuron}}$ are given by redefining $v_{ijlr}^{\alpha\beta}$ as $[\Phi^\top]_{i\alpha}[\Phi^\top]_{j\alpha}[\Phi^\top]_{l\beta}[\Phi^\top]_{r\beta}$, i.e. simply compute everything with the transposed data matrix.

All the estimators can be straightforwardly implemented using the einsum operation. We use the einsum function provided in JAX.

### A.4 NOISE CORRECTION DERIVATION

Suppose that for each stimulus $x_i$ and neuron $w_\alpha$, the observed activation in trial $t$ is

$$\Phi_{i\alpha}^{(t)} = \phi(x_i, w_\alpha) + \eta(x_i, w_\alpha, \epsilon_t),$$

where $\epsilon_t$ is sampled independently across trials, and $\mathbb{E}_\epsilon[\eta(x, w, \epsilon)] = 0$ for all $x$ and $w$. Additive noise corresponds to $\eta$ not depending on $\phi$; multiplicative noise corresponds to $\eta(x, w, \epsilon) = \epsilon \, \phi(x, w)$ with $\mathbb{E}[\epsilon] = 0$.

**Cross-product construction.** Given two independent trial matrices $\Phi^{(1)}$ and $\Phi^{(2)}$, redefine the elementary quartic tensor as

$$v_{ijkl}^{\alpha\beta} \leftarrow \Phi_{i\alpha}^{(1)} \Phi_{j\alpha}^{(2)} \Phi_{k\beta}^{(1)} \Phi_{l\beta}^{(2)}.$$

**Claim.** For indices $(i, j, k, l)$ that are all distinct and $\alpha \neq \beta$,

$$\mathbb{E}_{\epsilon_1, \epsilon_2}\left[v_{ijkl}^{\alpha\beta}\right] = \phi(x_i, w_\alpha) \, \phi(x_j, w_\alpha) \, \phi(x_k, w_\beta) \, \phi(x_l, w_\beta).$$

**Proof.** Since $\epsilon_1$ and $\epsilon_2$ are independent, the two factors $\Phi_{i\alpha}^{(1)} \Phi_{j\alpha}^{(2)}$ and $\Phi_{k\beta}^{(1)} \Phi_{l\beta}^{(2)}$ are also independent (they involve disjoint sets of noise realizations once conditioned on $\{x_i\}$ and $\{w_\alpha\}$). For the row factor:

$$\mathbb{E}\left[\Phi_{i\alpha}^{(1)} \Phi_{j\alpha}^{(2)}\right] = \mathbb{E}\left[(\phi_{i\alpha} + \eta_{i\alpha}^{(1)})(\phi_{j\alpha} + \eta_{j\alpha}^{(2)})\right]$$

$$= \phi_{i\alpha}\phi_{j\alpha} + \phi_{i\alpha} \underbrace{\mathbb{E}[\eta_{j\alpha}^{(2)}]}_{=0} + \underbrace{\mathbb{E}[\eta_{i\alpha}^{(1)}]}_{=0} \phi_{j\alpha} + \underbrace{\mathbb{E}[\eta_{i\alpha}^{(1)}]\mathbb{E}[\eta_{j\alpha}^{(2)}]}_{=0} = \phi_{i\alpha}\phi_{j\alpha},$$

where we used $\mathbb{E}[\eta^{(1)}] = 0$, $\mathbb{E}[\eta^{(2)}] = 0$, and independence of the two trials. The column factor gives the same result. Multiplying gives the claimed expectation, so the bias-corrected estimators $A_{\text{both}}$ and $B_{\text{both}}$ applied to the cross-product matrix are unbiased estimators of $A$ and $B$ from Eq. Eq. (3). The remaining bias is therefore purely of order $\mathcal{O}(1/P + 1/Q)$, arising from the unequal-index averaging.

**Correlated noise.** If $\epsilon_1$ and $\epsilon_2$ are *correlated across trials* (e.g., slow drift, gain modulation), then $\mathbb{E}[\eta^{(1)}\eta^{(2)}] \neq 0$, and the above cancellation is incomplete. Specifically, for $i = j$ (diagonal terms, which appear in $t_{\text{both}}^1$ and $t_{\text{both}}^3$),

$$\mathbb{E}\left[\Phi_{i\alpha}^{(1)}\Phi_{i\alpha}^{(2)}\right] = \phi_{i\alpha}^2 + \mathbb{E}\left[\eta_{i\alpha}^{(1)}\eta_{i\alpha}^{(2)}\right].$$

The residual bias from correlated noise contributes $\mathcal{O}(1/P + 1/Q)$ multiplied by a factor proportional to the cross-trial noise covariance. Using $N$ trials and averaging over all $N(N-1)/2$ cross-trial pairs reduces this contribution by a factor of $1/N$, i.e., to $\mathcal{O}(1/N)$.

**Summary of noise structures.** The cross-product construction eliminates bias contributed by *independent* additive and multiplicative noise — noise whose realizations $\epsilon_1, \epsilon_2$ are uncorrelated across the two trials — reducing the noise-induced bias to $\mathcal{O}(1/P + 1/Q)$. Noise that is *correlated across trials* (e.g., slow neural drift, gain modulations, or stimulus-locked hemodynamic variation in fMRI) is *not* eliminated: the residual bias is proportional to the cross-trial noise covariance $\mathbb{E}[\eta^{(1)}\eta^{(2)}]$. Using $N > 2$ trials and averaging all $N(N-1)/2$ cross-trial pairs reduces this correlated-noise contribution by a factor of $1/N$.

## A.5 IMPORTANCE SAMPLING

In practice, sampling distributions of stimuli and neurons may not match the intended distributions. For instance, electrophysiology probes tend to capture neurons from spatially clustered cortical layers or depths, and stimulus sets may be imbalanced across categories. These biases can introduce design-specific errors not removed by the unequal-index averaging alone.

We resolve this via importance sampling. Suppose the latent variables are drawn from the *true* distributions $x_i \sim \rho_{\mathcal{X}}$ and $w_\alpha \sim \rho_{\mathcal{W}}$, but the observed sample matrix is generated under biased (observed) distributions $x_i \sim \rho_{\mathcal{X}}^{\text{obs}}$ and $w_\alpha \sim \rho_{\mathcal{W}}^{\text{obs}}$, where the supports of $\rho_{\mathcal{X}}$ and $\rho_{\mathcal{W}}$ are subsets of those of $\rho_{\mathcal{X}}^{\text{obs}}$ and $\rho_{\mathcal{W}}^{\text{obs}}$, respectively. Define the importance weights

$$r(x) := \frac{\rho_{\mathcal{X}}(x)}{\rho_{\mathcal{X}}^{\text{obs}}(x)}, \qquad c(w) := \frac{\rho_{\mathcal{W}}(w)}{\rho_{\mathcal{W}}^{\text{obs}}(w)}.$$

We introduce a weighted unequal-index summation

$$\left\langle v_{ijlr}^{\alpha\beta} \right\rangle_{\text{both}}^{\text{IS}} := \frac{\sum_{i \neq j \neq l \neq r}^{\#r, \mathcal{X}_s} \sum_{\alpha \neq \beta}^{\#c, \mathcal{W}_s} r(x_i)r(x_j)r(x_l)r(x_r)\, c(w_\alpha)c(w_\beta)\, v_{ijlr}^{\alpha\beta}}{\# \text{ of summands}},$$

where the row sum iterates over the observed stimulus set $\mathcal{X}_s$ and the column sum iterates over the observed neuron set $\mathcal{W}_s$, each excluding colliding indices. The importance-sampling-corrected estimator is then

$$\gamma_{\text{both}}^{\text{IS}} := \frac{\left\langle v_{iijj}^{\alpha\beta} \right\rangle_{\text{both}}^{\text{IS}} - 2\left\langle v_{iijl}^{\alpha\beta} \right\rangle_{\text{both}}^{\text{IS}} + \left\langle v_{ijlr}^{\alpha\beta} \right\rangle_{\text{both}}^{\text{IS}}}{\left\langle v_{ijij}^{\alpha\beta} \right\rangle_{\text{both}}^{\text{IS}} - 2\left\langle v_{ijjl}^{\alpha\beta} \right\rangle_{\text{both}}^{\text{IS}} + \left\langle v_{ijlr}^{\alpha\beta} \right\rangle_{\text{both}}^{\text{IS}}},$$

whose numerator and denominator are unbiased estimators of those of $\gamma$. As a concrete example, if one aims to estimate the PR of V1 representations but recordings are biased toward neurons with specific tuning orientations, one can construct a histogram over tuning orientations to estimate $\rho_{\mathcal{W}}^{\text{obs}}$, assume a uniform true distribution $\rho_{\mathcal{W}}$, and compute $c(w)$ accordingly. Similarly, stimulus-set imbalances across object categories can be corrected by setting $r(x)$ to the inverse of the category sampling frequency.

## A.6 ESTIMATING THE PR OF A FINITE-SIZE UNDERLYING MATRIX

Suppose the sample matrix $\Phi \in \mathbb{R}^{P \times Q}$ is obtained by uniformly sampling the rows and columns of $G \in \mathbb{R}^{R \times C}$ without replacement. The goal is to estimate the PR of the matrix $G$ with column-wise centering:

$$\gamma = \frac{\bar{t}_g^1 - 2\bar{t}_g^2 + \bar{t}_g^5}{\bar{t}_g^3 - 2\bar{t}_g^4 + \bar{t}_g^5}$$

where

$$\bar{t}_g^1 := \langle g_{iijj}^{\alpha\beta} \rangle, \quad \bar{t}_g^2 := \langle g_{iijl}^{\alpha\beta} \rangle, \quad \bar{t}_g^3 := \langle g_{ijij}^{\alpha\beta} \rangle, \quad \bar{t}_g^4 := \langle g_{ijjl}^{\alpha\beta} \rangle, \quad \text{and} \quad \bar{t}_g^5 := \langle g_{ijlr}^{\alpha\beta} \rangle,$$

where $g_{ijlr}^{\alpha\beta} := G_{i\alpha} G_{j\alpha} G_{l\beta} G_{r\beta}$, and the bracket here means simply sum over the free indices.

The following are the unbiased estimators, i.e. $t_g^1, t_g^2, t_g^3, t_g^4,$ and $t_g^5$, of these terms respectively:

$$
\begin{aligned}
t_g^1 := \frac{1}{R^2 C^2} \Bigg( &\frac{R(R-1)\,C(C-1)}{P(P-1)\,Q(Q-1)} \sum_{i \neq j} \sum_{\alpha \neq \beta} v_{iijj}^{\alpha\beta} + \frac{R\,C(C-1)}{P\,Q(Q-1)} \sum_i \sum_{\alpha \neq \beta} v_{iiii}^{\alpha\beta} \\
&+ \frac{R(R-1)\,C}{P(P-1)\,Q} \sum_{i \neq j} \sum_\alpha v_{iijj}^{\alpha\alpha} + \frac{R\,C}{P\,Q} \sum_i \sum_\alpha v_{iiii}^{\alpha\alpha} \Bigg),
\end{aligned}
\tag{S11}
$$

$$
\begin{aligned}
t_g^2 := &\frac{1}{R^3 C^2} \Bigg( \frac{R(R-1)(R-2)\,C(C-1)}{P(P-1)(P-2)\,Q(Q-1)} \sum_{i \neq j \neq l} \sum_{\alpha \neq \beta} v_{iijl}^{\alpha\beta} \\
&+ \frac{R(R-1)\,C(C-1)}{P(P-1)\,Q(Q-1)} \sum_{i \neq j} \sum_{\alpha \neq \beta} (2\,v_{iiij}^{\alpha\beta} + v_{iijj}^{\alpha\beta}) + \frac{R\,C(C-1)}{P\,Q(Q-1)} \sum_i \sum_{\alpha \neq \beta} v_{iiii}^{\alpha\beta} \Bigg) \\
+ &\frac{1}{R^3 C^2} \Bigg( \frac{R(R-1)(R-2)\,C}{P(P-1)(P-2)\,Q} \sum_{i \neq j \neq l} \sum_\alpha v_{iijl}^{\alpha\alpha} \\
&+ \frac{R(R-1)\,C}{P(P-1)\,Q} \sum_{i \neq j} \sum_\alpha (2\,v_{iiij}^{\alpha\alpha} + v_{iijj}^{\alpha\alpha}) + \frac{R\,C}{P\,Q} \sum_i \sum_\alpha v_{iiii}^{\alpha\alpha} \Bigg),
\end{aligned}
\tag{S12}
$$

$$
\begin{aligned}
t_g^3 := \frac{1}{R^2 C^2} \Bigg( &\frac{R(R-1)\,C(C-1)}{P(P-1)\,Q(Q-1)} \sum_{i \neq j} \sum_{\alpha \neq \beta} v_{ijij}^{\alpha\beta} + \frac{R\,C(C-1)}{P\,Q(Q-1)} \sum_i \sum_{\alpha \neq \beta} v_{iiii}^{\alpha\beta} \\
&+ \frac{R(R-1)\,C}{P(P-1)\,Q} \sum_{i \neq j} \sum_\alpha v_{ijij}^{\alpha\alpha} + \frac{R\,C}{P\,Q} \sum_i \sum_\alpha v_{iiii}^{\alpha\alpha} \Bigg),
\end{aligned}
\tag{S13}
$$

$$
\begin{aligned}
t_g^4 := &\frac{1}{R^3 C^2} \Bigg( \frac{R(R-1)(R-2)\,C(C-1)}{P(P-1)(P-2)\,Q(Q-1)} \sum_{i \neq j \neq l} \sum_{\alpha \neq \beta} v_{ijjl}^{\alpha\beta} \\
&+ \frac{R(R-1)\,C(C-1)}{P(P-1)\,Q(Q-1)} \sum_{i \neq j} \sum_{\alpha \neq \beta} (2\,v_{iiij}^{\alpha\beta} + v_{ijij}^{\alpha\beta}) + \frac{R\,C(C-1)}{P\,Q(Q-1)} \sum_i \sum_{\alpha \neq \beta} v_{iiii}^{\alpha\beta} \Bigg) \\
+ &\frac{1}{R^3 C^2} \Bigg( \frac{R(R-1)(R-2)\,C}{P(P-1)(P-2)\,Q} \sum_{i \neq j \neq l} \sum_\alpha v_{ijjl}^{\alpha\alpha} \\
&+ \frac{R(R-1)\,C}{P(P-1)\,Q} \sum_{i \neq j} \sum_\alpha (2\,v_{iiij}^{\alpha\alpha} + v_{ijij}^{\alpha\alpha}) + \frac{R\,C}{P\,Q} \sum_i \sum_\alpha v_{iiii}^{\alpha\alpha} \Bigg),
\end{aligned}
\tag{S14}
$$

and

$$
\begin{aligned}
t_g^5 :=\ & \frac{1}{R^4 C^2}\Bigg( \frac{R(R{-}1)(R{-}2)(R{-}3)\,C(C{-}1)}{P(P{-}1)(P{-}2)(P{-}3)\,Q(Q{-}1)} \sum_{i\neq j\neq l\neq r}\sum_{\alpha\neq\beta} v_{ijlr}^{\alpha\beta} \\
& + \frac{R(R{-}1)(R{-}2)\,C(C{-}1)}{P(P{-}1)(P{-}2)\,Q(Q{-}1)} \sum_{i\neq j\neq l}\sum_{\alpha\neq\beta}\big(2\,v_{iijl}^{\alpha\beta}+4\,v_{ijjl}^{\alpha\beta}\big) \\
& + 4\,\frac{R(R{-}1)\,C(C{-}1)}{P(P{-}1)\,Q(Q{-}1)} \sum_{i\neq j}\sum_{\alpha\neq\beta} v_{iiij}^{\alpha\beta}\ +\ \frac{R\,C(C{-}1)}{P\,Q(Q{-}1)} \sum_{i}\sum_{\alpha\neq\beta} v_{iiii}^{\alpha\beta}\Bigg) \\
& + \frac{1}{R^4 C^2}\Bigg( \frac{R(R{-}1)(R{-}2)(R{-}3)\,C}{P(P{-}1)(P{-}2)(P{-}3)\,Q} \sum_{i\neq j\neq l\neq r}\sum_{\alpha} v_{ijlr}^{\alpha\alpha} \\
& + \frac{R(R{-}1)(R{-}2)\,C}{P(P{-}1)(P{-}2)\,Q} \sum_{i\neq j\neq l}\sum_{\alpha}\big(2\,v_{iijl}^{\alpha\alpha}+4\,v_{ijjl}^{\alpha\alpha}\big) \\
& + 4\,\frac{R(R{-}1)\,C}{P(P{-}1)\,Q} \sum_{i\neq j}\sum_{\alpha} v_{iiij}^{\alpha\alpha}\ +\ \frac{R\,C}{P\,Q} \sum_{i}\sum_{\alpha} v_{iiii}^{\alpha\alpha}\Bigg),
\end{aligned}
\tag{S15}
$$

where $v_{ijlr}^{\alpha\beta} := \Phi_{i\alpha}\,\Phi_{j\alpha}\,\Phi_{l\beta}\,\Phi_{r\beta}$.

To implement this function as a computer program, we need to vectorize the summations over unequal indices in an efficient manner. For that, we provide the following expansions of these recurring sums. Suppose

$$
r_{ijlr} := \frac{1}{Q(Q-1)} \sum_{\alpha\neq\beta} v_{ijlr}^{\alpha\beta} = \frac{1}{Q(Q-1)} \Bigg( \sum_{\alpha,\beta} v_{ijlr}^{\alpha\beta} - \sum_{\alpha} v_{ijlr}^{\alpha\alpha} \Bigg).
$$

Then

$$\frac{\sum_i \sum_{\alpha \neq \beta} v_{iiii}^{\alpha\beta}}{P\,Q(Q-1)} = \frac{\sum_i r_{iiii}}{P}, \tag{S16}$$

$$\frac{\sum_{i\neq j} \sum_{\alpha \neq \beta} v_{iijj}^{\alpha\beta}}{P(P-1)\,Q(Q-1)} = \frac{\sum_{i,j} r_{iijj} - \sum_i r_{iiii}}{P(P-1)}, \tag{S17}$$

$$\frac{\sum_{i\neq j} \sum_{\alpha \neq \beta} v_{iiij}^{\alpha\beta}}{P(P-1)\,Q(Q-1)} = \frac{\sum_{i,j} r_{iiij} - \sum_i r_{iiii}}{P(P-1)}, \tag{S18}$$

$$\frac{\sum_{i\neq j} \sum_{\alpha \neq \beta} v_{ijij}^{\alpha\beta}}{P(P-1)\,Q(Q-1)} = \frac{\sum_{i,j} r_{ijij} - \sum_i r_{iiii}}{P(P-1)}, \tag{S19}$$

$$\frac{\sum_{i\neq j\neq l} \sum_{\alpha \neq \beta} v_{iijl}^{\alpha\beta}}{P(P-1)(P-2)\,Q(Q-1)} = \frac{\sum_{i,j,l} r_{iijl} - 2\sum_{i,j} r_{iiij} - \sum_{i,j} r_{iijj} + 2\sum_i r_{iiii}}{P(P-1)(P-2)}, \tag{S20}$$

$$\frac{\sum_{i\neq j\neq l} \sum_{\alpha \neq \beta} v_{ijjl}^{\alpha\beta}}{P(P-1)(P-2)\,Q(Q-1)} = \frac{\sum_{i,j,l} r_{ijjl} - 2\sum_{i,j} r_{iiij} - \sum_{i,j} r_{ijij} + 2\sum_i r_{iiii}}{P(P-1)(P-2)}, \tag{S21}$$

$$\frac{\sum_{i\neq j\neq l\neq r} \sum_{\alpha \neq \beta} v_{ijlr}^{\alpha\beta}}{P(P-1)(P-2)(P-3)\,Q(Q-1)} = \frac{1}{P(P-1)(P-2)(P-3)}\left( \sum_{i,j,l,m} r_{ijlm} - 2\sum_{i,j,l}\left(r_{iijl} + 2\,r_{ijjl}\right) \right.$$
$$\left. + \sum_{i,j}\left(r_{iijj} + 8\,r_{ijjj} + 2\,r_{ijij}\right) - 6\sum_i r_{iiii} \right). \tag{S22}$$

In $t_g^1$, $t_g^2$, $t_g^3$, $t_g^4$, and $t_g^5$, we also have a version of each expression (one whole line) above, where we have $\sum_\alpha$ instead of $\sum_{\alpha\neq\beta}$. In that case, one can simply replace $r_{ijlr} = \frac{1}{Q(Q-1)}\left(\sum_{\alpha,\beta} v_{ijlr}^{\alpha\beta} - \sum_\alpha v_{ijlr}^{\alpha\alpha}\right)$ with $r''_{ijlr} = \frac{1}{Q}\sum_\alpha v_{ijlr}^{\alpha\alpha}$ and plug it into the R.H.S. of each expression above wherever $r$ appears.

### A.6.1  EXAMPLE DERIVATION

As an example, here we derive $t_g^3$, an estimator for $\bar{t}_g^3$, which is the denominator of the PR without the centering. This also corresponds to the second moment of the eigenvalues of $G$. The following derivation of $t_g^3$ should illuminate the rest of the derivation, which readers can easily deduce.

Let us start with the expression of what we want to estimate

$$\bar{t}_g^3 = \left\langle g_{ijij}^{\alpha\beta} \right\rangle = \frac{1}{R^2C^2} \sum_{i,j} \sum_{\alpha,\beta} g_{ijij}^{\alpha\beta}.$$

Now consider the following sum and its expansion:

$$\frac{1}{R^2 C^2} \sum_{i,j} \sum_{\alpha,\beta} g_{ijij}^{\alpha\beta} \equiv \frac{1}{R^2 C^2} \left( \sum_{i\neq j} \sum_{\alpha\neq\beta} g_{ijij}^{\alpha\beta} + \sum_{i} \sum_{\alpha\neq\beta} g_{iiii}^{\alpha\beta} + \sum_{i\neq j} \sum_{\alpha} g_{ijij}^{\alpha\alpha} + \sum_{i} \sum_{\alpha} g_{iiii}^{\alpha\alpha} \right).$$

The goal is to estimate each term without bias.

Let us try estimating $\sum_{i\neq j} \sum_{\alpha\neq\beta} g_{ijij}^{\alpha\beta}$ with the following estimator, which we derived earlier for the infinite underlying matrix case:

$$\frac{1}{P(P-1)Q(Q-1)} \sum_{i\neq j} \sum_{\alpha\neq\beta} v_{ijij}^{\alpha\beta}.$$

If we take the expected value of it, we have

$$\left\langle \frac{1}{P(P-1)Q(Q-1)} \sum_{i\neq j} \sum_{\alpha\neq\beta} \phi(x_i, w_\alpha)\phi(x_j, w_\alpha)\phi(x_i, w_\beta)\phi(x_j, w_\beta) \right\rangle$$

$$= \left\langle \phi(x,w)\phi(y,w)\phi(x,u)\phi(y,u) \right\rangle_{x,y,w,u}$$

$$= \left\langle x^\top Gw y^\top Gw x^\top Gu y^\top Gu \right\rangle_{x,y,w,u}$$

$$= \left\langle \sum_{ijkl} \sum_{\alpha\beta\theta\omega} \left( x^{(i)}y^{(j)}x^{(k)}y^{(l)} \right) \left( w^{(\alpha)}w^{(\beta)}u^{(\theta)}u^{(\omega)} \right) G_{i\alpha}G_{j\beta}G_{k\theta}G_{l\omega} \right\rangle_{x,y,w,u}$$

where $x$ is a vector and $x^{(i)}$ is a scalar value indicating the $i$th dimension of $x$ (similar for $y^{(j)}$, $w^{(\alpha)}$, and $u^{(\alpha)}$). Here, we defined the generative process as

$$\Phi_{i\alpha} = \phi(x_i, w_\alpha) = x_i^\top Gw_\alpha$$

where $x_i$ and $w_\alpha$ are randomly generated one-hot vectors. $x_i$ and $w_\alpha$ are drawn such that they select the rows and columns of $G$ without replacement. Therefore, unlike the original setup, $\{x_i\}_{i=1}^P$ are not independent, and so are $\{w_\alpha\}_{\alpha=1}^Q$.

Since $x$ is a one-hot vector, $x^{(i)}x^{(k)} = 0$, if $i \neq k$. The same logic applies to $y$, $w$, and $u$. Also note that $x^{(i)}y^{(j)} = 0$, if $i = j$, and $w^{(\alpha)}u^{(\theta)} = 0$, if $\alpha = \theta$. Therefore, the above simplifies to

$$\left\langle \frac{1}{P(P-1)Q(Q-1)} \sum_{i\neq j} \sum_{\alpha\neq\beta} v_{ijij}^{\alpha\beta} \right\rangle = \left\langle \sum_{i\neq j} \sum_{\alpha\neq\theta} \left( x^{(i)}y^{(j)} \right) \left( w^{(\alpha)}u^{(\theta)} \right) g_{ijij}^{\alpha\theta} \right\rangle_{x,y,w,u}$$

$$= \frac{1}{R(R-1)C(C-1)} \sum_{i\neq j} \sum_{\alpha\neq\theta} g_{ijij}^{\alpha\theta}$$

Therefore, we have estimated $\sum_{i\neq j} \sum_{\alpha\neq\theta} g_{ijij}^{\alpha\theta}$ but with the extra factor $\frac{1}{R(R-1)C(C-1)}$. Therefore, we know that the following is an unbiased estimator of $\sum_{i\neq j} \sum_{\alpha\neq\theta} g_{ijij}^{\alpha\theta}$:

$$\frac{R(R-1)C(C-1)}{P(P-1)Q(Q-1)} \sum_{i\neq j} \sum_{\alpha\neq\beta} v_{ijij}^{\alpha\beta}.$$

The estimators of $\sum_{i} \sum_{\alpha\neq\beta} g_{iiii}^{\alpha\beta}$, $\sum_{i\neq j} \sum_{\alpha} g_{ijij}^{\alpha\alpha}$, and $\sum_{i} \sum_{\alpha} g_{iiii}^{\alpha\alpha}$ can be derived in the same manner. The same logic applies to deriving all terms for the estimators of $t_g^1$, $t_g^2$, $t_g^4$, and $t_g^5$.

# B  APPENDIX FOR SCALING LAW OF THE NAIVE ESTIMATOR

Here, we quantify the expected value of the inverse of the naive estimator. In other words, we quantify the expected dimensionality of the sample matrix $\Phi \in \mathbb{R}^{P \times Q}$. For simplicity, we do not consider the centering, and assume $\gamma$ is large and $\left\langle \frac{1}{\gamma_{\text{naive}}} \right\rangle$ is close to $\frac{1}{\gamma}$.

We start with the second-order Taylor series expansion of the ratio of some random variables $h$ and $l$, i.e. $\frac{h}{l}$, around $\frac{H}{L}$, where $H$ and $L$ are the expected value of $h$ and $l$ respectively.

$$\left\langle \frac{h}{l} \right\rangle \to \frac{H}{L} \left( 1 + \frac{1}{H} \delta_h - \frac{1}{L} \delta_l - \frac{1}{HL} (\sigma_{hl} + \delta_h \delta_l) + \frac{1}{L^2} \left( \sigma_l^2 + \delta_l^2 \right) \right)$$

where $\delta_h := \langle h \rangle - H$, $\delta_l := \langle l \rangle - L$, $\sigma_{hl} = \text{Cov}(h, l)$, and $\sigma_l^2 = \text{Var}(l)$.

Since we quantify the expected value of the inverse of the naive estimator, $l$ corresponds to the numerator of $\gamma_{\text{naive}}$ and $h$ corresponds to the denominator of $\gamma_{\text{naive}}$, i.e.,

$$l = \frac{1}{P^2 Q^2} \sum_{i,j} \sum_{\alpha,\beta} \phi(x_i, w_\alpha) \phi(x_i, w_\alpha) \phi(x_j, w_\beta) \phi(x_j, w_\beta),$$

and

$$h = \frac{1}{P^2 Q^2} \sum_{i,j} \sum_{\alpha,\beta} \phi(x_i, w_\alpha) \phi(x_j, w_\alpha) \phi(x_i, w_\beta) \phi(x_j, w_\beta).$$

The indices run over the samples $\{x_i\}_{i=1}^P$ and $\{w_\alpha\}_{\alpha=1}^Q$ that are independently drawn from $\rho_\mathcal{X}$ and $\rho_\mathcal{W}$ respectively. After some algebra, we arrive at the following expressions for the quantities relevant for quantifying $\left\langle \frac{h}{l} \right\rangle$ above.

$$\delta_l = \frac{1}{P} \left( \left\langle k(x,x)^2 \right\rangle - \left\langle k(x,x) \right\rangle^2 \right) + \frac{1}{Q} \left( \left\langle \tilde{k}(w,w)^2 \right\rangle - \left\langle k(x,x) \right\rangle^2 \right),$$

$$\delta_h = \frac{1}{P} \left( \left\langle k(x,x)^2 \right\rangle - \left\langle k(x,y)^2 \right\rangle \right) + \frac{1}{Q} \left( \left\langle \tilde{k}(w,w)^2 \right\rangle - \left\langle k(x,y)^2 \right\rangle \right),$$

$$\sigma_{lh} = \frac{4}{P} \left( \left\langle k(x,y)^2 k(x,x) \right\rangle \left\langle k(x,x) \right\rangle - \left\langle k(x,y)^2 \right\rangle \left\langle k(x,x) \right\rangle^2 \right)$$
$$+ \frac{4}{Q} \left( \left\langle \tilde{k}(w,u)^2 \tilde{k}(w,w) \right\rangle \left\langle \tilde{k}(w,w) \right\rangle - \left\langle k(x,y)^2 \right\rangle \left\langle k(x,x) \right\rangle^2 \right),$$

$$\sigma_l^2 = -\frac{6}{P} \left\langle k(x,x) \right\rangle^4 + \frac{4}{P} \left\langle k(x,x)^2 \right\rangle \left\langle k(x,x) \right\rangle^2 + \frac{2}{P} \left\langle k(x,x) \right\rangle^2 \left\langle k(x,y)^2 \right\rangle$$
$$- \frac{6}{Q} \left\langle \tilde{k}(w,w) \right\rangle^4 + \frac{4}{Q} \left\langle \tilde{k}(w,w)^2 \right\rangle \left\langle \tilde{k}(w,w) \right\rangle^2 + \frac{2}{Q} \left\langle \tilde{k}(w,w) \right\rangle^2 \left\langle \tilde{k}(w,u)^2 \right\rangle,$$

and

$$\sigma_h^2 = \frac{4}{P} \left( \left\langle k(x,y)^2 k(y,z)^2 \right\rangle - \left\langle k(x,y)^2 \right\rangle^2 \right) + \frac{4}{Q} \left( \left\langle \tilde{k}(w,u)^2 \tilde{k}(u,v)^2 \right\rangle - \left\langle \tilde{k}(w,u)^2 \right\rangle^2 \right).$$

Plugging the above into the approximate formula for $\left\langle \frac{h}{l} \right\rangle$, we get

$$\left\langle \frac{1}{\gamma_{\text{naive}}} \right\rangle \to \frac{1}{\gamma} + \left( \frac{1}{P\psi} + \frac{1}{Q\tilde{\psi}} \right) \left( 1 - \frac{1}{\gamma} \right)$$

$$- \frac{4}{P\gamma} \left( \frac{\left\langle k(x,y)^2 k(x,x) \right\rangle}{\left\langle k(x,y)^2 \right\rangle \left\langle k(x,x) \right\rangle} - 1 \right) - \frac{4}{Q\gamma} \left( \frac{\left\langle \tilde{k}(w,u)^2 \tilde{k}(w,w) \right\rangle}{\left\langle \tilde{k}(w,u)^2 \right\rangle \left\langle \tilde{k}(w,w) \right\rangle} - 1 \right)$$

$$- 6\frac{1}{\gamma} \left( \frac{1}{P} + \frac{1}{Q} \right) + 4\frac{1}{\gamma} \left( \frac{1}{P\psi} + \frac{1}{Q\tilde{\psi}} \right) + \frac{2}{\gamma^2} \left( \frac{1}{P} + \frac{1}{Q} \right),$$

where $\psi := \frac{\left\langle k(x,x) \right\rangle^2}{\left\langle k(x,x)^2 \right\rangle}$ and $\tilde{\psi} := \frac{\left\langle \tilde{k}(w,w) \right\rangle^2}{\left\langle \tilde{k}(w,w)^2 \right\rangle}$, which are the participation ratios of $k(x,x)$ and $\tilde{k}(w,w)$ with respect to $\rho_{\mathcal{X}}$ and $\rho_{\mathcal{W}}$, respectively.

Now we take the large $\gamma$ assumption, which simplifies the above to

$$\left\langle \frac{1}{\gamma_{\text{naive}}} \right\rangle \to \frac{1}{\gamma} + \frac{1}{P\psi} + \frac{1}{Q\tilde{\psi}}.$$

Intuitively, $\psi$ is the effective fraction of stimuli that evokes large overall response in the neural population (i.e., the effective fraction of rows with large norms), and $\tilde{\psi}$ is the effective fraction of neurons that has large overall response to the stimuli (i.e., the effective fraction of columns with large norms). If we assume $k(x,x)$ and $\tilde{k}(w,w)$ are constants ($\psi = \tilde{\psi} = 1$), then we have

$$\left\langle \frac{1}{\gamma_{\text{naive}}} \right\rangle \to \frac{1}{\gamma} + \frac{1}{P} + \frac{1}{Q}.$$

A large class of generative process has $k(x,x)$ and $\tilde{k}(w,w)$ that are constants. For example, consider the linear process where sampling of $\Phi$ is modeled as a random projection of some fixed ground truth matrix $G \in \mathbb{R}^{R \times C}$,

$$\Phi_{i,\alpha} = x_i^\top G w_\alpha,$$

where $x_i \sim \mathcal{N}(0, I_{R \times R})$ and $w_\alpha \sim \mathcal{N}(0, I_{C \times C})$, independently across $i$ and $\alpha$. This process is widely used to approximate a matrix sampling in physics and random matrix theories, and has $\psi = \tilde{\psi} = 1$.

## C   EXPERIMENTAL DETAILS

We provide all our code and necessary data to produce the figures with our supplementary material. Here, we provide additional results from our experiments.

### C.1   NEURAL DATA EXPERIMENTS

In all our experiments, we computed the dimensionality of subsampled activations over many iterations (`Stringer`: 50, `MajajHong`: 500, `TVSD`: 250, and `ThingsFMRI`: 50).

Here, we show that the bias in our estimator arises from the nonlinear nature of the division operation used in computing dimensionality. Note that the dimensionality is defined as the ratio between $A$ and $B$ as in Eq. (3). In Fig. S1, the first row shows our estimator when dimensionality is calculated by averaging the ratio, $\left\langle \frac{A}{B} \right\rangle$, while the second row shows the ratio of the averaged numerator and denominator, $\frac{\langle A \rangle}{\langle B \rangle}$. While our estimator gives unbiased estimators of $A$ and $B$ separately, it remains biased for dimensionality.

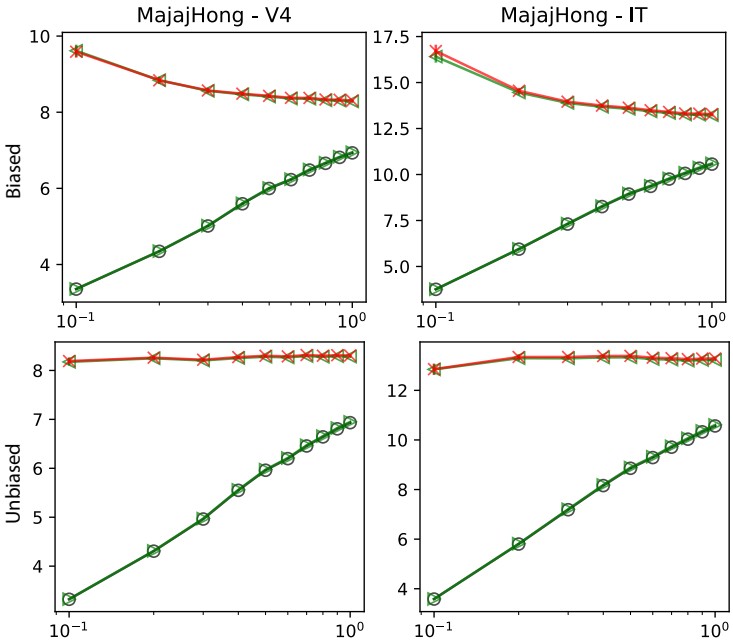

Figure S1: Bias due to nonlinearity in the definition of dimensionality.

## C.2 LLM EXPERIMENTS

For our experiments, we use the pretrained Llama 3 base model from Grattafiori et al. (2024). We extract its hidden representations on nine different languages from the Flores dataset NLLB Team et al. (2024): `English`, `French`, `Japanese`, `Korean`, `Russian`, `Ukranian`, `Turkish`, `Kazakh`, and `Greek`.

## D BIAS VARIANCE ANALYSIS

Here we quantify the bias and variance of the $\gamma_{\text{naive}}$ and $\gamma_{\text{both}}$, in the context where both the rows and columns are sampled. $P$ and $Q$ are the number of sampled rows and columns, respectively. We will ignore the centering operation for simplicity. We start by considering the general behavior of the ratio of estimators.

Suppose $A$ is an unbiased estimate of $a$ and $B$ is an unbiased estimate of $b$. Suppose $\delta_A$ and $\delta_B$ are the biases of $A$ and $B$, respectively, e.g. $\delta_A = \langle A \rangle - a$, $\sigma_A^2$ and $\sigma_B^2$ are the variances of $A$ and $B$, respectively, and $\sigma_{AB}$ is the covariance of $A$ and $B$. Then the following are the bias and variance of $A/B$ as an estimator of $a/b$, up to the second-order approximations:

$$\text{bias}\left(\frac{A}{B}\right) \approx \frac{a}{b}\left(\frac{1}{a}\delta_A - \frac{1}{b}\delta_B - \frac{1}{ab}\left(\sigma_{AB} + \delta_A\delta_B\right) + \frac{1}{b^2}\left(\sigma_B^2 + \delta_B^2\right)\right)$$

$$\text{var}\left(\frac{A}{B}\right) \approx \frac{a^2}{b^2}\left(\frac{1}{a^2}\sigma_A^2 + \frac{1}{b^2}\sigma_B^2 - 2\frac{1}{ab}\sigma_{AB}\right)$$

In our case, $a = \langle k(x,x)\rangle^2$ and $b = \langle k(x,y)^2\rangle$

$$\text{bias}\left(\frac{A}{B}\right) \approx \gamma\left(\frac{\delta_A}{\langle k(x,x)\rangle^2} - \frac{\delta_B}{\langle k(x,y)^2\rangle} - \frac{\sigma_{AB} + \delta_A\delta_B}{\langle k(x,x)\rangle^2 \langle k(x,y)^2\rangle} + \frac{\sigma_B^2 + \delta_B^2}{\langle k(x,y)^2\rangle^2}\right)$$

$$\mathrm{var}\left(\frac{A}{B}\right) \approx \gamma^2 \left( \frac{\sigma_A^2}{\langle k(x,x)\rangle^4} + \frac{\sigma_B^2}{\langle k(x,y)^2\rangle^2} - 2\frac{\sigma_{AB}}{\langle k(x,x)\rangle^2 \langle k(x,y)^2\rangle} \right)$$

## D.1 NAIVE ESTIMATOR

We have the following for the naive estimator $\gamma_{\mathrm{naive}}$:

$$\delta_A = \frac{1}{P}\left( \left\langle k(x,x)^2 \right\rangle - \left\langle k(x,x)\right\rangle^2 \right) + \frac{1}{Q}\left( \left\langle \tilde{k}(w,w)^2 \right\rangle - \left\langle k(x,x)\right\rangle^2 \right)$$

$$\delta_B = \frac{1}{P}\left( \left\langle k(x,x)^2 \right\rangle - \left\langle k(x,y)^2 \right\rangle \right) + \frac{1}{Q}\left( \left\langle \tilde{k}(w,w)^2 \right\rangle - \left\langle k(x,y)^2 \right\rangle \right)$$

$$\sigma_{AB} = \frac{4}{P}\left( \left\langle k(x,y)^2 k(x,x) \right\rangle \langle k(x,x)\rangle - \left\langle k(x,y)^2 \right\rangle \langle k(x,x)\rangle^2 \right) +$$
$$\frac{4}{Q}\left( \left\langle \tilde{k}(w,u)^2 \tilde{k}(w,w) \right\rangle \langle \tilde{k}(w,w)\rangle - \left\langle k(x,y)^2 \right\rangle \langle k(x,x)\rangle^2 \right)$$

$$\sigma_A^2 = -\frac{6}{P}\langle k(x,x)\rangle^4 + \frac{4}{P}\left\langle k(x,x)^2 \right\rangle \langle k(x,x)\rangle^2 + \frac{2}{P}\langle k(x,x)\rangle^2 \left\langle k(x,y)^2 \right\rangle$$
$$- \frac{6}{Q}\left\langle \tilde{k}(w,w) \right\rangle^4 + \frac{4}{Q}\left\langle \tilde{k}(w,w)^2 \right\rangle \left\langle \tilde{k}(w,w)\right\rangle^2 + \frac{2}{Q}\left\langle \tilde{k}(w,w)\right\rangle^2 \left\langle \tilde{k}(w,u)^2 \right\rangle$$

$$\sigma_B^2 = \frac{4}{P}\left( \left\langle k(x,y)^2 k(y,z)^2 \right\rangle - \left\langle k(x,y)^2 \right\rangle^2 \right) + \frac{4}{Q}\left( \left\langle \tilde{k}(w,u)^2 \tilde{k}(u,v)^2 \right\rangle - \left\langle \tilde{k}(w,u)^2 \right\rangle^2 \right)$$

Plugging the above into the earlier expressions, the bias and variance of the naive estimator are given by

$$\mathrm{bias}\left(\gamma_{\mathrm{naive}}\right) \approx 4\gamma \left( \frac{1}{P}(c - c') + \frac{1}{Q}(\tilde{c} - \tilde{c}') \right) - \gamma(\gamma - 1)\left( \frac{1}{P\psi} + \frac{1}{Q\tilde{\psi}} \right), \quad \text{and}$$

$$\mathrm{var}\left(\gamma_{\mathrm{naive}}\right) \approx 4\frac{\gamma^2}{P}\left( \frac{1}{\psi} + c - 2c' \right) + 4\frac{\gamma^2}{Q}\left( \frac{1}{\tilde{\psi}} + \tilde{c} - 2\tilde{c}' \right) - 2\gamma(\gamma - 1)\left( \frac{1}{P} + \frac{1}{Q} \right)$$

where

$$c := \frac{\left\langle \langle k(x,y)^2\rangle_x^2 \right\rangle_y}{\langle k(x,y)^2\rangle_{x,y}^2}, \quad c' := \frac{\langle k(x,y)^2 k(x,x)\rangle}{\langle k(x,y)^2\rangle \langle k(x,x)\rangle},$$

$$\tilde{c} := \frac{\left\langle \langle \tilde{k}(w,u)^2\rangle_w^2 \right\rangle_u}{\langle \tilde{k}(w,u)^2\rangle_{w,u}^2}, \quad \tilde{c}' := \frac{\langle \tilde{k}(w,u)^2 \tilde{k}(w,w)\rangle}{\langle \tilde{k}(w,u)^2\rangle \langle \tilde{k}(w,w)\rangle},$$

$$\psi := \frac{\langle k(x,x)\rangle^2}{\langle k(x,x)^2\rangle}, \quad \text{and} \quad \tilde{\psi} := \frac{\langle \tilde{k}(w,w)\rangle^2}{\langle \tilde{k}(w,w)^2\rangle}.$$

## D.2 OUR ESTIMATOR

We have the following for our estimator $\gamma_{\text{both}}$:

$$\delta_A = 0$$

$$\delta_B = 0$$

$$\sigma_{AB} = \frac{4}{P} \left( \left\langle k(x,y)^2 k(x,x) \right\rangle \left\langle k(x,x) \right\rangle - \left\langle k(x,y)^2 \right\rangle \left\langle k(x,x) \right\rangle^2 \right)$$
$$+ \frac{4}{Q} \left( \left\langle \tilde{k}(w,u)^2 \tilde{k}(w,w) \right\rangle \left\langle \tilde{k}(w,w) \right\rangle - \left\langle k(x,y)^2 \right\rangle \left\langle k(x,x) \right\rangle^2 \right)$$

$$\sigma_A^2 = \frac{4}{P} \left( \left\langle k(x,x)^2 \right\rangle \left\langle k(x,x) \right\rangle^2 - \left\langle k(x,x) \right\rangle^4 \right) + \frac{4}{Q} \left( \left\langle \tilde{k}(w,w)^2 \right\rangle \left\langle \tilde{k}(w,w) \right\rangle^2 - \left\langle k(x,x) \right\rangle^4 \right)$$

$$\sigma_B^2 = \frac{4}{P} \left( \left\langle k(x,y)^2 k(y,z)^2 \right\rangle - \left\langle k(x,y)^2 \right\rangle^2 \right) + \frac{4}{Q} \left( \left\langle \tilde{k}(w,u)^2 \tilde{k}(u,v)^2 \right\rangle - \left\langle k(x,y)^2 \right\rangle^2 \right)$$

Plugging the above into the earlier expressions, the bias and variance of our estimator are given by

$$\text{bias} \left( \gamma_{\text{both}} \right) \approx 4\gamma \left( \frac{1}{P} \left( c - c' \right) + \frac{1}{Q} \left( \tilde{c} - \tilde{c}' \right) \right), \quad \text{and}$$

$$\text{var} \left( \gamma_{\text{both}} \right) \approx 4 \frac{\gamma^2}{P} \left( \frac{1}{\psi} + c - 2c' \right) + 4 \frac{\gamma^2}{Q} \left( \frac{1}{\tilde{\psi}} + \tilde{c} - 2\tilde{c}' \right).$$

## D.3 COMPARISON OF THE BIAS OF $\gamma_{\text{NAIVE}}$ AND $\gamma_{\text{BOTH}}$

In summary, we have the following for the biases:

$$\text{bias} \left( \gamma_{\text{naive}} \right) = 4\gamma \left( \frac{1}{P} \left( c - c' \right) + \frac{1}{Q} \left( \tilde{c} - \tilde{c}' \right) \right) - \gamma \left( \gamma - 1 \right) \left( \frac{1}{P\psi} + \frac{1}{Q\tilde{\psi}} \right) + \mathcal{O} \left( \left( \frac{1}{P} + \frac{1}{Q} \right)^2 \right),$$

$$\text{and} \quad \text{bias} \left( \gamma_{\text{both}} \right) = 4\gamma \left( \frac{1}{P} \left( c - c' \right) + \frac{1}{Q} \left( \tilde{c} - \tilde{c}' \right) \right) + \mathcal{O} \left( \left( \frac{1}{P} + \frac{1}{Q} \right)^2 \right)$$

Note that the first terms $4\gamma \left( \frac{1}{P} \left( c - c' \right) + \frac{1}{Q} \left( \tilde{c} - \tilde{c}' \right) \right)$ in $\text{bias} \left( \gamma_{\text{naive}} \right)$ is entirely the leading order term of $\text{bias} \left( \gamma_{\text{both}} \right)$. This means that the second term $\gamma \left( \gamma - 1 \right) \left( \frac{1}{P\psi} + \frac{1}{Q\tilde{\psi}} \right)$ is the additional bias in $\gamma_{\text{naive}}$. Here we show that the first term is often small, and the second term tends to be large, which is removed in $\text{bias} \left( \gamma_{\text{both}} \right)$. Let us inspect the term $c - c'$ that contributed to the bias in $\gamma_{\text{both}}$:

$$c - c' = \frac{\left\langle \left\langle k(x,y)^2 \right\rangle_x^2 \right\rangle_y}{\left\langle k(x,y)^2 \right\rangle_{x,y}^2} - \frac{\left\langle k(x,y)^2 k(x,x) \right\rangle_{x,y}}{\left\langle k(x,y)^2 \right\rangle_{x,y} \left\langle k(x,x) \right\rangle_x},$$

If $\left\langle k(x,y)^2 \right\rangle_x$ is constant for all $y$, then $c = 1$. This is also when $c' = 1$. Therefore, when $\left\langle k(x,y)^2 \right\rangle_x$ is constant,

$$c - c' = 0.$$

Note that $\langle k(x,y)^2 \rangle_{x \sim \rho_X}$ is constant for all $y$, if the kernel and data distribution are matched by a global symmetry (rotational or translational). For example, consider any dot-product kernel of the form

$$k(x,y) = h\,(x \cdot y)$$

where $h : \mathbb{R} \to \mathbb{R}$, with $x$ and $y$ sampled uniformly from a unit sphere. Then $c - c' = 0$, so the bias in our estimator is dramatically smaller than that of the naive estimator. Similarly, consider any translation-invariant kernel of the form

$$k(x,y) = h(x - y)$$

with $x$ and $y$ sampled from a translation-invariant distribution, e.g. uniform measure on a torus. Then $c - c' = 0$.

# E   LOCAL DIMENSIONALITY ESTIMATION

The sample neural manifold is simply the set of row vectors of the sample matrix $\Phi$:

$$\mathcal{M} := \left\{ \bar{\phi}_i \right\}_{i=1}^{P}$$

where $\bar{\phi}_i \in \mathbb{R}^Q$ is the $i$th row vector of $\Phi$. Suppose the distance metric defined in the sample representation space $\mathbb{R}^Q$ is denoted as $d\,(\cdot, \cdot)$. Define the ball in the representation space centered around $\bar{\phi}_0 \in \mathbb{R}^Q$ with radius $r$:

$$\mathcal{B}(\bar{\phi}_0, r) := \left\{ \bar{\phi} \in \mathcal{M} \mid d(\bar{\phi}, \bar{\phi}_0) \leq r \right\}.$$

Let $\gamma_{\text{both}}\,(\mathcal{S})$ denote a dimensionality measured on a finite set $\mathcal{S}$ of sample representations in $\mathbb{R}^Q$. Then, the local dimensionality estimate for a radius $r$ is defined as

$$\gamma_{\text{both}}^{\text{local}}(r) = \frac{1}{|\mathcal{M}|} \sum_{\bar{\phi}_0 \in \mathcal{M}} \gamma_{\text{both}}\left( \mathcal{B}(\bar{\phi}_0, r) \right).$$

Now, let us define the distance metric. For two representation vectors $\bar{\phi}, \bar{\phi}_0 \in \mathbb{R}^Q$, the squared Mahalanobis distance is defined as:

$$d^2(\bar{\phi}, \bar{\phi}_0) = \left( \bar{\phi} - \bar{\phi}_0 \right)^{\top} M \left( \bar{\phi} - \bar{\phi}_0 \right)$$

where $M$ is a positive-definite metric. We want to preferably select representation along the tangent directions of $\bar{\phi}_0$ so that we faithfully capture the local dimensionality. We estimate a local representation metric by $\Sigma\left(\bar{\phi}_0\right)$, the covariance of the $k$-nearest neighbors of $\bar{\phi}_0$. Then we define $M$ as $\Sigma\left(\bar{\phi}_0\right)^{\dagger}$, the pseudoinverse of $\Sigma\left(\bar{\phi}_0\right)$.

