# OpenReview forum: "Estimating Dimensionality of Neural Representations from Finite Samples"
_ICLR.cc/2026/Conference — ICLR 2026 Poster_

### Official Review · Reviewer_uXtA · 2025-10-21

**Soundness:** 4
**Presentation:** 3
**Contribution:** 4
**Rating:** 8
**Confidence:** 3

**Summary:**

This submission proposes a new estimator of the Participation Ratio (PR), a well-known tool to measure the global dimensionality in Machine Learning or Neuroscience. This novel estimator is proven to be less biased and more robust to measurement noise than the state-of-the-art. The main impact is that it needs fewer measurements to yield a precise estimate.

**Strengths:**

I see the following strengths:
- The topic fits exactly the scope of ICLR, ie. learning representation. This new estimator is helpful in gauging the 'quality' of a learned representation.
- The paper is well written, postponing the gory technical details in the appendices. At first sight, reading the main body, I had the impression that the exposition was too simple, but the appendices evidence that the rationale is strong. I suggest to add more cross-references to the appendices.
- The approach is deductive: first an analysis of the biases in the naive approach, then the deduction of the modifications to patch this caveat.
- The new estimator is not entirely unbiased, but the study shows that the bias is significantly smaller
- This main property is also demonstrated with convincing experiments.
- The computation of the new estimator is expensive. Yet, the submission proposes an efficient implementation.
- The submission is dense as it also proposes two extensions: robustness to measurement noise with a double trial, and an integration of a weighting mechanism to estimate local dimensionalities, which can then be averaged into a global dimensionality.

**Weaknesses:**

- W1: The experimental protocol only compares the new estimator to the naive PR estimate and Two-NN. There are many more estimators based on nearest neighbours, like Gride from Denti et al. (cited in the submission), the Hill estimator for local dimensionality and the likes (MLE, MoM, TLE, see the works of Michael Houle, James Bailey -- not cited in the submission), not to mentioned the `insane' estimator based on diffusion models (see A Geometric View of Data Complexity: Efficient Local Intrinsic Dimension Estimation with Diffusion Models, Kamkari et al., NeurIPS 2024).

- W2: I expected some justifications about the noise correction based on double trials. On the same token, I do not understand how appendix B is related to the main body.

- W3: I am not sure that the dimensionality estimator should be the arithmetic mean of local estimates. Some works argue that the harmonic mean is better. See for instance, page 11 of *A Survey of Dimension Estimation Methods*, Binnie et al.

**Questions:**

- Suggestions: Adding pointers to the appendices in the main body will make the submission sharper. I also suggest numbering more equations to allow these precise cross-references. Another option would be to formulate results as Propositions whose proofs are given in the appendix... but I can live without this formalism.

-  Q1: Fig. 1. What is the value of Q when P is varying (left)?  What is the value of $P$ when $Q$ is varying (right)?

- Q2: Why $\gamma_{col}$ ($\gamma_{row}$) is a scaled copy of $\gamma_{both}$ when $P$ (resp. $Q$) varies?

- Q3: I am not convinced by the use of Mahalanobis distance to define the nearest neighbours. Typically, NN-based Local dimensionality estimators use plain Euclidean distance. What is the value of k for learning the local covariance matrix? You mention 4 neighbours in the main body.

Typos:
- Line 760: "Consider $\hat{t}^5$..." but the following details the comptutation of $t^1$.
- Line 907: Similarly

---

> ### Author Response · Authors · 2025-12-03
>
> We thank the reviewer for the positive and constructive feedback.
>
> - I expected some justifications about the noise correction based on double trials. On the same token, I do not understand how appendix B is related to the main body.
>
> Thank you for pointing these out. We will include details regarding the noise correction in the supplementary material. We will remove Supplementary section B, as it is not relevant to the main text.
>
> - Fig. 1. What is the value of Q when P is varying (left)? What is the value of $P$ when $Q$ is varying (right)?
>
> We will include the values of P and Q in Figure 1.
>
> - Why $\gamma_{col}$ ($\gamma_{row}$) is a scaled copy of $\gamma_{both}$ when $P$ (resp. $Q$) varies?
>
> That is not necessarily true. Both are insensitive to $P$ (hence the flatline), but $\gamma_{row}$ is biased due to finite $Q$, whereas $\gamma_{both}$ is insensitive to both $P$ and $Q$.
>
> - I am not convinced by the use of Mahalanobis distance to define the nearest neighbours. Typically, NN-based Local dimensionality estimators use plain Euclidean distance. What is the value of k for learning the local covariance matrix? You mention 4 neighbours in the main body.
>
> 4 is the minimum number of neighbors ($k$) we require. Instead of fixing $k$, we fix the radius of the local ball, allowing $k$ to vary. We use the Mahalanobis distance to account for the manifold's local curvature and anisotropy, ensuring that the selected neighbors are statistically relevant to the local covariance structure, rather than just spatially close in Euclidean space.

---

### Official Review · Reviewer_aVtZ · 2025-10-30

**Soundness:** 3
**Presentation:** 3
**Contribution:** 3
**Rating:** 6
**Confidence:** 4

**Summary:**

The paper studies estimates of global manifold dimensionality from finite neural response matrices. It shows that the standard participation ratio (PR) is systematically biased in small-sample regimes and proposes a bias-corrected PR estimator that removes terms arising from index collisions in finite submatrices. Concretely, the sample matrix is modeled as a random submatrix of an underlying infinite matrix, obtained by independently and uniformly sampling stimuli and neurons. The authors derive unbiased estimators for the numerator and denominator of PR by averaging only over unequal indices and define γ_both as their ratio, with row-only and column-only variants (γ_row, γ_col). They further give a two-trial noise correction that cancels additive or multiplicative noise without requiring many repeats, and a weighted version that yields a local, noise-robust intrinsic dimensionality when weights select neighbors. Experiments on synthetic data recover ground-truth dimensionality, and applications to mouse V1 calcium imaging, macaque electrophysiology, human IT fMRI, and LLM activations show reduced dependence on P and Q compared to naive PR and TwoNN. Implementational tricks for unequal-index summations are provided.

**Strengths:**

* Originality: Clear identification of finite-sample bias sources in PR and a general correction via unequal-index averaging. Weighted and two-trial extensions broaden utility
* Quality: Solid derivations, sensible baselines, coverage across multiple data modalities
* Clarity: Motivations, assumptions, and estimator definitions are well laid out; limitations are stated.
* Significance: If adopted, this can standardize dimensionality reporting and reduce misleading sample-size effects across empirical work.

**Weaknesses:**

* The estimator assumes independent, uniform sampling of rows and columns. In practice, electrophysiology probes capture spatially clustered units and often bias toward specific cortical layers or depths; calcium imaging has field-of-view and sparsity biases; fMRI voxels are not neuron samples at all. This undermines the "random submatrix of the infinite one" assumption and could introduce design-specific bias not removed by unequal-index averaging. Please analyze or simulate non-uniform, block-dependent, or layer-biased sampling and show robustness or an adjusted estimator that incorporates known inclusion probabilities via importance weights
* The two-trial correction addresses additive or global multiplicative noise across trials. Real recordings can have correlated, heteroskedastic noise across neurons/time. Clarify which noise structures are neutralized by the cross-trial construction and which remain.
* No variance estimates or confidence intervals are given for γ_both. A delta-method approximation for a ratio of U-statistics or a simple bootstrap over rows and columns would help readers gauge reliability.
* The local estimator requires radius sweeps and unequal-index summations. Please report time and memory as functions of P and Q
* No comparisons to other global measures used in practice (under finite samples) are made. No ablations included that vary manifold curvature, anisotropy, and rank-plus-tail spectra

**Questions:**

* How does γ_both behave under non-uniform sampling where inclusion probabilities differ by layer or receptive field location. Can your weighted estimator correct such sampling bias if weights are set to inverse inclusion probabilities?
* The two-trial noise correction assumes independence of trial noise. How sensitive is γ_both to trial-to-trial drift or slow fluctuations that are shared across trials?
* For the LLM study, can you report layerwise γ with and without centering and with language-balanced sampling to ensure that row sampling is not conflated with sentence-length distribution?
* The supplement mentions different centered kernels for task vs neuron centering. Can you clarify in the main text when each is appropriate?

---

> ### Author Response · Authors · 2025-12-03
> **Correcting a sampling bias using importance sampling**
>
> We thank the reviewer for the valuable feedback. Here, we provide the complete solution to the problem of sampling bias that the reviewer raised. This will be included in the final draft of our paper.
>
> - “Please analyze or simulate non-uniform, block-dependent, or layer-biased sampling and show robustness or an adjusted estimator that incorporates known inclusion probabilities via importance weights” “Can your weighted estimator correct such sampling bias if weights are set to inverse inclusion probabilities?”
>
> First, we would like to clarify what it means to draw rows and columns uniformly from a matrix whose number of rows and columns approaches infinity. As stated in the paper, the sample matrix is generated by
>
> $$
> \Phi_{i,\alpha}=\phi(x_i,w_\alpha), \quad x_i\sim \rho_\mathcal{X},\, \quad w_\alpha \sim \rho_\mathcal{W}
> $$
>
> where $x_i$ and $w_\alpha$ are some latent variables independently drawn from some arbitrary distributions $\rho_\mathcal{X}$ and $\rho_\mathcal{W}$. The goal is to estimate the PR of a hypothetical matrix ($\Phi^{(\infty)}$) of infinite size generated from this generative process, from a finite-size sample matrix $\Phi$ generated from the same process. A roughly equivalent, yet non-rigorous statement is that $\Phi$ is a random submatrix of $\Phi^{(\infty)}$ generated by selecting the rows and columns of the latter uniformly at random. Note that $\rho_\mathcal{X}$ and $\rho_\mathcal{W}$ need not be a uniform distribution for this statement to be true.
>
> Now, in this framework, the reviewer's question can be written as: can we estimate the PR of $\Phi^{(\infty)}$ generated with $\rho_\mathcal{X}$  and $\rho_\mathcal{W}$ when the sample matrix $\Phi$ is generated by the following process instead:
>
> $$
> \Phi_{i,\alpha}=\phi(x_i,w_\alpha), \quad x_i\sim \rho_\mathcal{X}^{\text{obs}},\, \quad w_\alpha \sim \rho_\mathcal{W}^{\text{obs}}
> $$
>
> We assume the supports of  $\rho_\mathcal{X}$  and $\rho_\mathcal{W}$ are subsets of the supports of  $\rho_\mathcal{X}^{\text{obs}}$  and $\rho_\mathcal{W}^{\text{obs}}$, respectively. Ideally, $\rho_\mathcal{X}^{\text{obs}}$  and $\rho_\mathcal{W}^{\text{obs}}$ are identical to  $\rho_\mathcal{X}$  and $\rho_\mathcal{W}$, but in real life, the sampling of neurons and stimuli (inputs) can be biased, which can be modeled as $\rho_\mathcal{X} \neq \rho_\mathcal{X}^{\text{obs}}$  and $\rho_\mathcal{W} \neq \rho_\mathcal{W}^{\text{obs}}$.
>
>  We can resolve this issue by the method of importance sampling. To this end, we introduce yet another summation method described by
>
> $$
> \left\langle v_{ijlr}^{\alpha\beta}\right\rangle_{\text{both}}^{\text{IS}} = \frac{1}{\text{\\# of summands}} \times
> $$
>
> $$
> \sum_{ijlr}^{\\#r,\mathcal{X_s}}\sum_{\alpha \beta}^{\\#c,\mathcal{W_s}} v_{ijlr}^{\alpha\beta}
> $$
>
>
>
>
> where we are using the following notation
>
> $$
> \sum_{i_1,\ldots,i_n}^{\\#f,\mathcal{S}}T_{i_{1},\ldots,i_{n}} \equiv \sum_{i_{1},\ldots,i_{n}}^{\\#}\left[\prod_{l=1}^{n}f(x_{i_{l}})\right]T_{i_{1},\ldots,i_{n}}
> $$
>
>
> where $T$ is a tensor, $n$ is the number of the free indices being iterated over, and each index iterates over the elements of the ordered set $\mathcal{X_s}$,
> i.e. $i_l \in [ 1,\ldots,|\mathcal{X_s}| ]$, but skips when the index value matches that of any other index. We use the symbol $\sum_{i,j,k}^{\\#}$ to indicate the sum that excludes the instances where any of the indices are equal. $f:\mathcal{S}\to\mathbb{R}$ is an importance function. In our case, we require that the importance functions $r$ and $c$ take the following form:
>
> $$
> r(x)=\frac{\rho_{\mathcal{X}}(x)}{\rho_{\mathcal{X}}^{\text{obs}}(x)},\:\text{and}\quad c(w)=\frac{\rho_{\mathcal{W}}(w)}{\rho_{\mathcal{W}}^{\text{obs}}(w)}.
> $$
>
> Finally, the sampling bias-corrected estimator is
>
> $$
> \gamma_{\text{both}}^{\text{IS}} = \frac{\left\langle v_{iijj}^{\alpha\beta}\right\rangle_{\text{both}}^{\text{IS}}-2\left\langle v_{iijl}^{\alpha\beta}\right\rangle_{\text{both}}^{\text{IS}}+\left\langle v_{ijlr}^{\alpha\beta}\right\rangle_{\text{both}}^{\text{IS}}}{\left\langle v_{ijij}^{\alpha\beta}\right\rangle_{\text{both}}^{\text{IS}}-2\left\langle v_{ijjl}^{\alpha\beta}\right\rangle_{\text{both}}^{\text{IS}}+\left\langle v_{ijlr}^{\alpha\beta}\right\rangle_{\text{both}}^{\text{IS}}}
> $$
>
> whose denominators and numerators are unbiased estimators of those of $\gamma$. An example use case is when one aims to understand the PR of V1 representation, but the recording was biased towards neurons with specific tuning orientation due to experimental limitations. By constructing a histogram over the tuning orientations, one can estimate $\rho_{\mathcal{W}}^{\text{obs}}$, and by assuming uniform distribution for the true orientations $\rho_{\mathcal{W}}$, one can estimate $c$. Similarly, one can correct the sampling bias in the stimulus set, e.g. an imbalance in object categories in visual stimuli.

---

> ### Author Response · Authors · 2025-12-03
>
> - “The two-trial correction addresses additive or global multiplicative noise across trials. Real recordings can have correlated, heteroskedastic noise across neurons/time. Clarify which noise structures are neutralized by the cross-trial construction and which remain.” “The two-trial noise correction assumes independence of trial noise. How sensitive is γ_both to trial-to-trial drift or slow fluctuations that are shared across trials?”
>
> With two trials, the additive or multiplicative noise that is injected independently across rows and columns can be removed with the bias of order  $\mathcal{O}(1/P+1/Q)$. However, the effect of noise that is correlated across rows and columns would fall by only $1/2$. A simple extension to $N$ trials would make the contribution of the correlated noise fall like $\mathcal{O}(1/N)$. We will clarify this in the final draft.
>
> - No variance estimates or confidence intervals are given for γ_both. A delta-method approximation for a ratio of U-statistics or a simple bootstrap over rows and columns would help readers gauge reliability.
>
> We will include the confidence interval in the final draft.
>
> - No comparisons to other global measures used in practice (under finite samples) are made. No ablations included that vary manifold curvature, anisotropy, and rank-plus-tail spectra
>
> The current draft includes the comparisons to the naive PR estimator that is widely used in practice. We could include more synthetic data (such as the ones suggested by the reviewer), but we believe our theoretical analysis on the bias and variance of our estimator (Supplementary section D) provides a more holistic and concrete picture of how the properties of the kernel, input distribution, and parameter distributions affect our estimator's performance.
>
> - For the LLM study, can you report layerwise γ with and without centering and with language-balanced sampling to ensure that row sampling is not conflated with sentence-length distribution?
>
> We will include this in the final draft of the supplement.
>
> - The local estimator requires radius sweeps and unequal-index summations. Please report time and memory as functions of P and Q
>
> Following the notations in our paper, $P$ denotes the number of sample inputs (rows) and $Q$ denotes the number of sample features (columns). For computing the local PR estimate, the time complexity is dominated by the computation of the pairwise distance: $\mathcal{O}(P^2Q)$. The memory complexity is  $\mathcal{O}(P^2+PQ)$, since the sample matrix is of size $PQ$ and the distance matrix of size $P^2$. If we sweep through $r$ number of radii, then the time complexity is  $\mathcal{O}(rP^2Q)$ and the memory complexity is $\mathcal{O}(rP(P+Q))$,
>
> - The supplement mentions different centered kernels for task vs neuron centering. Can you clarify in the main text when each is appropriate?
>
> In the main text, we discuss the task-centering of a kernel. We will clarify this in the final draft of the supplementary section.

---

### Official Review · Reviewer_kPsS · 2025-10-31

**Soundness:** 3
**Presentation:** 3
**Contribution:** 3
**Rating:** 8
**Confidence:** 4

**Summary:**

The paper examines the problem of estimating the manifold dimensionality (effective rank) formed from neural data under stimuli, how current approaches may be biased especially in small sample sizes, and a new bias-corrected estimator. The authors demonstrate this metric on diverse neural data and a LLM.

**Strengths:**

Very well written and organized. It tackles an important and broad problem in computational neuroscience with some applications to general ML models. Math appears correct and well derived.

**Weaknesses:**

- Some jumps in the flow (e.g. how do line 266 and line 268 connect?) detracted from the clarity.

- Some portions of the results are unclear or unexplained. Figure 2 shows how some estimators don't change as a function of the number of samples included. But some of the naive and other estimators are stable, though the difference in value of dimensionality is different. Which is correct, given no ground truth? (See Q, top right and bottom left).

- Discussion and Limitations are too short; how feasible is this? How computationally expensive is it in practice?

**Questions:**

- How does this apply without inputs (e.g., spontaneous activity)?

- Is treating the sample activation matrix as a random submatrix always appropriate? Neurons are certainly subsampled, but in many organisms much more of the network is directly observable. If we observe active neurons, is it likely to be a random subsample, or simple most neurons that are responsive to a certain input stimulus? And most samples on the stimulus input side are not randomly chosen either.

- How much does the bias term contribute to the difference in estimators? E.g., how biased is biased? Line 312 denotes some bias as negligible without giving a value or order of magnitude. Line 358 seems to imply that 'most bias' means most change across number of samples. Figure 4 indicates that the new estimator is also biased?

---

> ### Author Response · Authors · 2025-12-03
>
> We thank the reviewer for the positive and constructive feedback.
>
> - Some jumps in the flow (e.g. how do line 266 and line 268 connect?) detracted from the clarity.
>
> We will improve the clarity in that section.
>
> - Some portions of the results are unclear or unexplained. Figure 2 shows how some estimators don't change as a function of the number of samples included. But some of the naive and other estimators are stable, though the difference in value of dimensionality is different. Which is correct, given no ground truth? (See Q, top right and bottom left).
>
> All estimators, except our $\gamma_{\text{both}}$ estimator, are sensitive to either the change of the number of rows (P) or columns (Q) or both, as shown in Figure 2. One of our estimators $\gamma_{\text{col}}$ is not sensitive to the number of columns (Q) but is sensitive to the number of rows (P). $\gamma_{\text{row}}$ exhibits the opposite behavior, whereas $\gamma_{\text{both}}$ is insensitive to both.
>
> - Discussion and Limitations are too short; how feasible is this? How computationally expensive is it in practice?
>
> The complexities of the naive and our estimators are the same:
> $O(\min(P, Q)^2 \cdot \max(P, Q))$. Both estimators can be computed quickly in practice.
>
> - How does this apply without inputs (e.g., spontaneous activity)?
>
> In the case where the temporal correlation of the sample neural population activity can be ignored, one can apply our estimator for estimating the dimensionality of the spontaneous activities.
>
> - Is treating the sample activation matrix as a random submatrix always appropriate? Neurons are certainly subsampled, but in many organisms much more of the network is directly observable. If we observe active neurons, is it likely to be a random subsample, or simple most neurons that are responsive to a certain input stimulus? And most samples on the stimulus input side are not randomly chosen either.
>
> We address this concern in our response to the Reviewer aVtZ. We use the method of importance sampling to resolve this potential problem in real-life data. We will include this in the final draft.
>
> - How much does the bias term contribute to the difference in estimators? E.g., how biased is biased? Line 312 denotes some bias as negligible without giving a value or order of magnitude. Line 358 seems to imply that 'most bias' means most change across number of samples.
>
> These questions are addressed in detail in the Supplementary section D (Bias variance analysis).
>
> - Figure 4 indicates that the new estimator is also biased?
>
> Figure 4 is for the local dimensionality (as opposed to the global dimensionality, which is the main focus of our work). The x-axis of Figure 4 is not the sample size, but the radius for the local dimensionality estimator. In the small radius limit, our local estimator converges to the true local dimensionality.

---

### Meta-Review · Area_Chair_r441 · 2025-12-14

**Summary:**

This paper estimates the intrinsic dimensionality for neural representations. In particular, the authors propose a bias-correction in participation-ratio (PR) estimators and extends to local and noise-robust dimensionality estimation. All reviewers recognize it as a simple, principled approach with theoretical grounding and empirical validations in neural brain recordings. The reviewers are concerned on the sampling/noise assumptions and missing uncertainty quantification. The authors resolve the concerns with importance sampling, and clarifications on modeling limitations, and commitment to include the confidence intervals. Overall, the merits outweigh the weaknesses, and I recommend acceptance.

**Reviewer Concerns:**

Reviewer kPsS is concerned on the clarity and limitations. Reviewer aVtZ is concerned on the sampling assumptions. Reviewer uXtA is concerned on limited empirical results. Most of these concerns are addressed in the rebuttal or clarified as limitations.

**Reviewer Scores:**

All reviewers are already on the (strong) acceptance side. From the review/rebuttal, they would not decrease their scores.

---

### Decision · Program_Chairs · 2026-01-26

Accept (Poster)